# CountTRuCoLa: Rule Learning for Interpretable Temporal Knowledge Graph Forecasting

## Abstract

We address the task of temporal knowledge graph (TKG) forecasting by introducing a fully interpretable method based on temporal rules. Motivated by recent work proposing a strong baseline using recurrent facts, our approach learns four simple types of rules with a confidence function that considers both recency and frequency. Evaluated on nine datasets, our method achieves performance that is competitive with state-of-the-art models and consistently outperforms the majority of them, while providing fully interpretable predictions.

## 1 Introduction

Temporal knowledge graphs (TKG) extend static knowledge graphs with temporal information by using timestamps that indicate when a triple is valid (Han et al., 2021a). Their ability to represent evolving multi-relational data enables reasoning over changing knowledge in a variety of tasks, including temporal question answering (Saxena et al., 2021), knowledge graph completion (Cai et al., 2023), and fact validation (Soulard et al., 2025). In recent years, the task of TKG forecasting, i.e. predicting future links in TKG, has attracted significant interest, leading to the development of diverse approaches (Jin et al., 2020; Li et al., 2022a;b). Examples for domain-specific use cases can be found in finance (Li et al., 2022b), career trajectory prediction (Lee et al., 2025), and clinical prediction (Sun et al., 2025).

This paper is motivated by a TKG forecasting baseline recently introduced by Gastinger et al. (2024b), which predicts future links purely based on the recurrence of facts. The authors demonstrate that despite its inability to capture complex dependencies, this baseline offers a competitive or even superior alternative to existing models on three out of five common datasets. These results challenge the capabilities of current models, which are often built on neural networks or use reinforcement learning architectures within a complex end-to-end architecture. Motivated by these considerations, we propose a deliberately simple and fully interpretable method for TKG forecasting, which picks up some ideas from the baseline proposed in Gastinger et al. (2024b) and extends it to a rule-based TKG forecasting model.

Our model comprises four types of rules, which cover simple frequency distributions of entities and relations as well as temporal regularities expressed as symbolic rules. Each of these temporal rules is associated with a confidence function, which assigns a score to a prediction based on two features: the temporal distance to the most current relevant observation and the frequency of the occurrences of the relevant observation. Our temporal rules use only a single body atom, meaning that the prediction of a rule is caused by a single fact and its occurrence frequency.

Within this paper we present a comprehensive experimental study comparing our method to nine existing approaches and two baselines. Our results show that the proposed model achieves competitive performance across diverse datasets, outperforming the majority of existing approaches in terms of mean reciprocal rank. Across all nine evaluation datasets, our method consistently ranks among the top performers, with the added benefits of being lightweight, fully interpretable, and computationally efficient.

## 2 RELATED WORK

Temporal knowledge graph forecasting has been addressed using a variety of approaches. Methods based on deep graph networks combine message-passing with sequential models to capture both structural and temporal patterns; notable examples include RE-Net (Jin et al., 2020), RE-GCN (Li et al., 2021b), xERTE (Han et al., 2021a), TANGO (Han et al., 2021b), RETIA (Liu et al., 2023a), and CEN (Li et al., 2022b). Another line of work leverages reinforcement learning agents to discover temporal paths, such as CluSTeR (Li et al., 2021a) and TimeTraveler (Sun et al., 2021). Other approaches include history-based prediction in CyGNet (Zhu et al., 2021), the use of local and global encoders in TiRGN (Li et al., 2022a), contrastive learning in CENET (Xu et al., 2023), latent relation mining in L2TKG (Zhang et al., 2023), and temporal path encoding in TPAR (Chen et al., 2024). More recently, LLM-based models such as zrLLM (Ding et al., 2024) and GenTKG (Liao et al., 2024) have been proposed to incorporate semantic information into TKG forecasting.

In addition, rule-based approaches for TKG forecasting aim to learn interpretable temporal logic rules. TLogic learns rules via temporal random walks (Liu et al., 2022), while TRKG extends this by introducing acyclic and relaxed-time-constraint rules (Kiran et al., 2023). TR-Rules further extends TLogic by supporting acyclic rules and introducing a window confidence measure to address temporal redundancy (Li et al., 2023). TempValid (Huang et al., 2024) builds on TLogic rules but explicitly models temporal validity by treating confidence and decay coefficients as learnable parameters within a machine learning framework, applying an exponential decay transformation to temporal features and aggregating them linearly. While the implementation details are not fully specified, TempValid likely relies on a neural or similar differentiable model, which, together with advanced negative sampling strategies, improves predictive performance at the expense of interpretability. Further, several approaches combine rules with embedding-based methods—such as ALRE-IR (Mei et al., 2022), LogE-Net (Liu et al., 2023b), TECHS (Lin et al., 2023), INFER (Li et al., 2025), and CognTKE (Chen et al., 2025). These hybrid methods generally sacrifice interpretability due to their reliance on learned embeddings and neural components.

Lastly, two deterministic heuristic baselines have been introduced: EdgeBank (Poursafaei et al., 2022), developed for single-relational graphs, does not consider relation types or temporal differences and simply predicts the recurrence of the same subjects and objects. In contrast, the baseline named Recurrency Baseline proposed by Gastinger et al. (2024b) predicts fact recurrence by incorporating the temporal distance and frequency of previous facts, and unlike EdgeBank, is relation-aware, modeling recurrence patterns separately for each relation type.

**Positioning of our Work**   Among these, the works closest to ours are TLogic and TempValid. However, both approaches have a different language bias that allows them to learn rules of length greater than one, that is, rules with multiple atoms in the body. While this increases expressiveness, it also introduces noise and reduces interpretability. In contrast, our approach intentionally focuses on the simplest and most interpretable form: rules with a single body atom. However, we do not simply reduce complexity. Instead, we design these short rules to be richer and more nuanced. Specifically, we introduce additional rule types beyond those in TLogic and TempValid, namely rules with constants, and rules reflecting general distributions inherent in the dataset, thus supporting a broader set of temporal patterns. We also propose an improved temporal scoring function with learned parameters, enabling more accurate modeling of temporal dynamics. In this context, our method not only takes into account the time distance to the most recent occurrence of the rule body but, in contrast to TempValid and TLogic, also the frequency of occurrence. By simplifying the rule form while refining its predictive power, our method offers an interpretable and effective alternative for TKG forecasting. Finally, while our work is inspired by the Recurrency Baseline, it goes beyond simple recurrence by introducing additional rule types that learn dependencies across different relations and by employing a dedicated parametrized confidence function for each rule.

## 3 BACKGROUND AND NOTATION

A temporal knowledge graph (TKG) $G$ is a set of quadruples $(s, p, o, t)$. We refer to $C(G) = \{s \mid (s, p, o, t) \in G\} \cup \{o \mid (s, p, o, t) \in G\}$ as entities (or constants) of $G$, $P(G) = \{p \mid (s, p, o, t) \in G\}$ as relations (or predicates) of $G$, $T(G) = \{t \mid (s, p, o, t) \in G\} \subset \mathbb{N}^+$ as timestamps of $G$. Timestamps may represent hours, days, years, or any other temporal granularity, depending on the

dataset and use case. The semantic meaning of a quadruple $(s, p, o, t)$ is that $s$ is in relation $p$ to $o$ at time $t$. Thus, a quadruple can also be understood as a triple $(s, p, o)$ that is additionally annotated with a timestamp $t$, which tells that the statement made by the triple is true at (or during) $t$.

Given a TKG $G$, TKG forecasting or extrapolation is the task of predicting quadruples $(s, p, o, t^\star)$ for future timestamps $t^\star > max(T(G))$ with $p \in P(G)$ and $s, o \in C(G)$. In this work we focus on the task of entity forecasting, that is, predicting object or subject entities for queries $(s, p, ?, t^\star)$ or $(?, p, o, t^\star)$. This task has become a common choice for measuring the predictive quality of TKG forecasting methods (Gastinger et al., 2023). Akin to static knowledge graph completion, TKG forecasting is approached as a ranking task (Han, 2022). For a given query, which is usually derived from a quadruple in a test or validation set, a model needs to rank entities in $C(G)$ using a scoring function that assigns plausibility scores. Due to that specific setting, for each query, there is always a correct entity to predict.

To simplify the evaluation protocol, one can extend a TKG $G$ by adding for each $(s, p, o, t) \in G$ the inverse quadruple $(o, p^{-1}, s, t)$ where $p^{-1} \notin P(G)$ denotes a fresh relation used as inverse relation of $p$. This doubles the size of $G$ and the number of relations. It allows to focus on object queries only by converting a subject query as $(?, p, o, t^\star)$ into the equivalent object query $(o, p^{-1}, ?, t^\star)$. We will see in the following section that it also reduces the number of required rules. Thus, we always extend a given TKG by its inverse quadruples. Note that most evaluation datasets are already available in that extended form.

We propose a model that learns logical rules to capture the temporal regularities inherent in $G$. In this context we use a prefix notation and understand relation $p$ as a ternary predicate. We use a literal $p(s, o, t)$ as logical representation of a quadruple $(s, p, o, t)$. A logical rule is a disjunction of literals with at most one unnegated literal, for example, $l_0 \vee \neg l_1 \vee \ldots \vee \neg l_n$. Rules are usually written in the equivalent implicative form $l_0 \leftarrow l_1 \wedge \ldots \wedge l_n$. We call $l_0$ the head of the rule and $l_1 \wedge \ldots \wedge l_n$ the body of the rule. Given a rule $r$, we use $\mathfrak{h}(r)$ to refer to its head and $\mathfrak{b}(r)$ to refer to its body.

Rules have already been applied successfully to static knowledge graph completion (KGC) tasks (Rossi et al., 2021). The following rule might have been used in this context.

$$workplace(x, y) \leftarrow livesIn(x, y) \tag{1}$$

It says that someone who lives in a certain city, will also work in that city with a certain probability. In the non-temporal setting the confidence value of a rule determines (or is aggregated into) the plausibility score which defines the position of a candidate within a ranking for a given query. The confidence of a rule $r$, which uses, for example, two variables $x$ and $y$ in its head and body, is defined, according to Galárraga et al. (2013), as follows:

$$conf_r = \frac{|\{(c, d) \mid \theta_{x/c, y/d}(\mathfrak{h}(r) \wedge \mathfrak{b}(r)) \in G\}|}{|\{(c, d) \mid \theta_{x/c, y/d}(\mathfrak{b}(r)) \in G\}| + \mathcal{P}}$$

where $\theta_{x/c, y/d}$ denotes a substitution of the variables $x$ and $y$ by constants $c$ and $d$ from $C(G)$. Thus, the numerator counts all possible substitutions to replace variables $x$ and $y$ such that the resulting groundings of both head and body are true according to $G$. The denominator is the sum of counting the larger set of those groundings where the body only is true (the head might or might not be true) and a constant $\mathcal{P}$. If we ignore $\mathcal{P}$, the confidence value estimates the probability that $r$ makes a correct prediction given $G$. $\mathcal{P}$, usually set to a small positive value, has been added to the denominator by Meilicke et al. (2024) as a smoothing parameter that pushes the confidence score towards 0 if the number of groundings is low. While static KGC works with static confidences, computing confidence values in the temporal case is less straight forward and requires to consider distances of timestamps of body and head groundings as input to confidence functions.

## 4 APPROACH

We explain how our approach finds and applies different rules, and how it learns temporal rule confidence functions such as shown in Figure 1, by counting occurrences of temporal triples. We refer to our approach as CountTRuCoLa (Temporal Rule Confidence Learning).

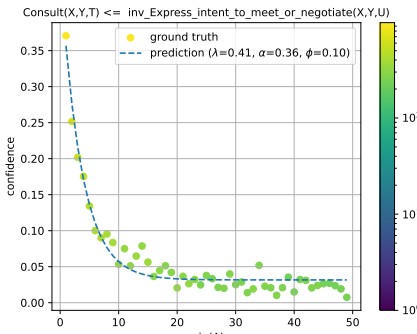 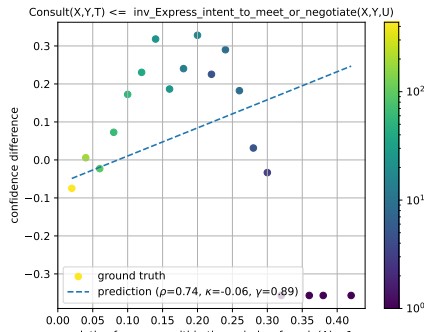

Figure 1: Examples $E_r$ (points) and predicted curves (blue lines) for one rule $r$. Colors indicate the number of samples in $E_r$. Left: Learning $f_r$, confidence vs. $\min(\Delta)$. Right: Learning $g_r$, residuals $\tilde{y} - f(\Delta)$ vs. frequency $\frac{|\Delta_W|}{W}$ for $\min(\Delta) = 1$.

### 4.1 RULES

Our approach supports four types of rules. The most important, called $xy$-rule, is shown in (2).

$$h(x, y, t^*) \leftarrow b(x, y, t) \ \wedge \ t^* > t \tag{2}$$

Aside from the constraint $t^* > t$, which enforces that we predict into the future, $xy$-rules contain only a single logical body atom. We call $xy$-rules where $h = b$ recurrent $xy$-rules. These recurrent $xy$-rules correspond to the core element in the baseline from Gastinger et al. (2024b). Rules (3) and (4) are examples for recurrent and non-recurrent $xy$-rules.

$$dated(x, y, t^*) \leftarrow dated(x, y, t) \ \wedge \ t^* > t \tag{3}$$

$$engaged(x, y, t^*) \leftarrow dated(x, y, t) \ \wedge \ t^* > t \tag{4}$$

Note that our approach does not support rules where the order of variables are flipped. We mentioned above that we extend each TKG by its inverse quadruples. For that reason a rule as $h(x, y, t^*) \leftarrow b(y, x, t)$ is equivalent to $h(x, y, t^*) \leftarrow b^{-1}(x, y, t)$.

Suppose we are concerned with a query $h(c, ?, t^\star)$. Relevant $xy$-rules are those rules that use $h$ in their head, for example, a rule $r = h(x, y, t^*) \leftarrow b(x, y, t) \wedge t^* > t$. Rule $r$ predicts a candidate $d$, i.e., $r$ predicts the quadruple $h(c, d, t^\star)$, if there exists a $t \in T(G)$ such that $b(c, d, t) \in G$. We can check this for a given $d$ via a constant time look-up by using appropriate index structures and we can retrieve all predicted candidates in linear time with respect to the number of candidates.

With respect to Rule (3), it makes obviously a difference if $x$ dated $y$ two weeks ago, or two years ago. Moreover, it might also be important to know how often $x$ and $y$ have been on dates within a certain time span. Thus, the confidence function that we introduce later on makes use of all time stamps $t'$ with $b(c, d, t') \in G$ to estimate the confidence of $h(c, d, t^\star)$. More precisely, it uses the distances between all these timestamps and $t^\star$. We introduce $\Delta^{r,G,t^\star}_{\theta_{x/c,y/d}}$ to refer to the set of these distances for the prediction $h(c, d, t^\star)$ made by rule $r$:

$$\Delta^{r,G,t^\star}_{\theta_{x/c,y/d}} = \{t^\star - t' \mid \theta_{x/c,y/d,t/t'}(\mathbf{b}(r)) \in G, t' \in T(G)\}$$

We use subscript $\theta_{x/c,y/d}$ to express that we are concerned with time distances for the prediction that we get if we substitute $x$ by $c$ and $y$ by $d$ in the body of $r$. We indexed $G$ to retrieve for each triple $(s, r, o)$ the set of timestamps $\{t \mid (s, r, o, t) \in G\}$ at which the triple has been true in constant time. We use such a look-up to construct the set of distances $\Delta^{r,G,t^\star}_{\theta_{x/c,y/d}}$, which is the input to the confidence function that we introduce below.

The second type of rule that we support, is a rule type that uses constants $d$ and $d'$ in head and body of the rule. We call an instance of the following rule type a $c$-rule.

$$h(x, d, t^*) \leftarrow b(x, d', t) \ \wedge \ t^* > t \tag{5}$$

The constants that appear in c-rules are usually frequent constants. The following example of a $c$-rule says that a person born in Amsterdam will (probably) study at the University of Amsterdam.

$$studied(x, uva, t^*) \leftarrow born(x, amsterdam, t) \ \wedge \ t^* > t \tag{6}$$

Similar to the $xy$-rule, a $c$-rule (5) predicts a candidate $d$, i.e., predicts the quadruple $h(c, d, t^\star)$, if there exists $t' \in T(G)$ such that $b(c, d', t') \in G$. Compared to the $xy$-rule, the set of distances relevant for the confidence computation depends only on the $x$-substitution, which means that $\Delta_{\theta_{x/c}}^{r,G,t^\star}$ is the relevant set given a $c$-rule $r$. A complete description requires a distinction between two directions. Please consult Appendix A.3 for details.

We include two further types of rules that use a static confidence, similar to the confidence value shown in Section 3. These rules can also be understood as a means to capture very basic frequency distributions within the dataset. We call these rules $z$-rules (7) and $f$-rules (8).

$$p(x, d, t) \leftarrow \exists z \; p(x, z, t) \tag{7}$$
$$p(c, d, t) \leftarrow \exists z \; p(c, z, t) \tag{8}$$

Both rules types, contrary to $xy$-rules and $c$-rules use the same timestamp $t$ in the head and in the body. Due to the evaluation protocol explained above, if we are asked a query $p(s, ?, t^\star)$, we are justified to assume that there exists a correct answer, i.e. that we can assume $\exists z \; p(s, z, t^\star)$, which corresponds to the body of a $z$-rule and a $f$-rule.

The following two examples illustrate the difference between these rule types.

$$eats(x, pizza, t) \leftarrow \exists z \; eats(x, z, t) \tag{9}$$
$$eats(kim, pizza, t) \leftarrow \exists z \; eats(kim, z, t) \tag{10}$$

The confidence value of (9) corresponds to the probability that someone eats pizza, if that person eats something. The confidence value of (10) corresponds to the probability that Kim eats pizza, if she eats something. Depending on Kim's eating habits, these numbers might diverge significantly. We compute the confidences for $z$ and $f$-rules following the definition in Section 3. However, we count not only over constants but over combinations of constants and timestamps.

Finally, we have to clarify that $xy$-rules and $c$-rules have also an additional existentially quantified atom similar to the body atom of the $z$-rule and $f$-rule, namely the atom $\exists z \; h(x, z, t^*)$. If we are asked a query $h(s, ?, t^\star)$ we are justified to assume that this atom is true due to the fact that the query is asked. Remember that each query has always at least one correct answer. Thus, the additional atom makes no difference when answering a query, but makes a difference when collecting the examples to learn the confidence function from.

We construct each possible $xy$-rule ($|P(G)|^2$ combinations), each $z$-rule ($|P(G)| \times |C(G)|$ combinations) and each $f$-rule ($|P(G)| \times |C(G)|^2$ combinations). These numbers are an upper bound. The actual numbers are much lower, as certain combinations of constants and relations do not appear together. However, the combinatorial space is too large for mining all $c$-rules. For that reason we collect a subset of all $c$-rules that make use of frequent constants only.

### 4.2 TEMPORAL CONFIDENCE FUNCTIONS

Each $xy$- and $c$-rule is associated with a parameterized temporal confidence function. To learn the parameters for a rule $r$, we collect examples $E_r$, representing the positive-to-negative ratio at different time distances and frequencies. Such examples are illustrated by colored points in Figure 1. The collection procedure is explained below, with pseudocode in Appendix A.1. Note that all steps rely exclusively on information from the training set. We first introduce a notation $G^t = \{(s, p, o, t') | (s, p, o, t') \in G \wedge t' < t\}$ that allows us to refer to the subset of $G$ that contains all quadruples up to a certain timestamp $t$. The procedure for collecting examples $E_r$ for an $xy$-rule $r = h(x, y, t^*) \leftarrow b(x, y, t)$ is organized in two nested loops. The outer loop iterates over all timestamps in $t \in T(G)$. This allows us to view $G^{t-1}$ as known observations, which enable us to make predictions for a future timestamp $t$. Within the outer loop, we have a nested inner loop that iterates over all pairs $(c, d)$ with $c, d \in C(G)$. For each combination it checks if there exists some $z$ with $h(c, z, t) \in G$, ensuring that only entities that might actually appear in a query are included in the example set. Then we collect the time distances from the $\theta_{x/c, y/d}$-substitution if applied to $r$ for making a prediction that targets timestamp $t$. Note that we collect only distances within a time window $\mathcal{W}$, where $\mathcal{W}$ is a hyperparameter. If the set is not empty we proceed as follows: If the prediction is correct, i.e., $h(c, d, t) \in G$, we store the set of distances as a positive example. If the prediction is incorrect, we store it as negative example. Then we continue with the next $(c, d)$ pair.

For a $c$-rule, the procedure is the same with two minor modifications. We can omit to loop over $d$ and, as there is no variable $y$ in the rule body, we have to drop $y/d$ from the substitution $\theta_{x/c,y/d}$.

To compute the confidence of a $xy$ or $c$-rule with respect to the time differences between body and head, we use $\Delta_{\theta_{x/c,y/d}}^{r,G,t^*}$, which we denote by $\Delta$ for simplicity. Our confidence function for a rule $r$ is defined by:

$$conf_r(\Delta) = f_r(\Delta) + g_r(\Delta) \tag{11}$$

The first summand, $f_r$, describes the effect of the most recent occurrence, i.e., the smallest time distance $\min(\Delta)$, of the body grounding:

$$f_r(\Delta) = \alpha_r \cdot \frac{1}{1 + \phi_r} \cdot (2^{-\lambda_r \cdot (\min(\Delta) - 1)} + \phi_r) \tag{12}$$

where $\alpha_r, \lambda_r, \phi_r \in \mathbb{R}_0^+$ are learnable parameters.

The second summand, $g_r$, captures the effect of the frequency, i.e., the number of occurrences $|\Delta|$, of the body grounding:

$$g_r(\Delta) = \min(\max(\rho_r \frac{|\Delta_{\mathcal{W}}|}{\mathcal{W}} + \frac{\kappa_r}{\min(\Delta)}, -\gamma_r), \gamma_r) \tag{13}$$

where $\Delta_{\mathcal{W}}$ denotes the set of body groundings whose time difference is at most $\mathcal{W}$, i.e., $\Delta_{\mathcal{W}} = \{d \in \Delta : d \leq w\}$, with $\mathcal{W}$ being a hyperparameter. Further, $\rho_r, \kappa_r, \gamma_r \in \mathbb{R}$ are learnable parameters. Note that the additive offset in $g_r$ depends inversely on $\min(\Delta)$. Intuitively, when $\min(\Delta)$ is low, a high proportion of the window $\mathcal{W}$ can contain body groundings. In contrast, when $\min(\Delta)$ is high, the maximum possible number of additional groundings in $\Delta_{\mathcal{W}}$ is reduced, since the upper limit for $|\Delta_{\mathcal{W}}|$ becomes $w - min(\Delta) + 1$.

Each parameter in the confidence function has a specific and interpretable role:

- $\alpha_r$: scales the $f_r$ score and determines its value when $\min(\Delta) = 1$.
- $\lambda_r$: specifies the decay rate, determining how quickly $f_r$ decreases as $\min(\Delta)$ increases. When $\lambda_r = 0$, the time distance between body and head has no effect while higher values result in faster decay.
- $\phi_r$: determines the asymptotic lower bound of $f_r$ as $\min(\Delta) \to \infty$.
- $\rho_r$: defines the slope of the frequency term $g_r$. When $\rho = 0$, frequency has no impact; for larger $\rho_r$, the score increases proportionally with observed frequency.
- $\kappa_r$: acts as an additive offset modulated by $\min(\Delta)$.
- $\gamma_r$: bounds frequency score via clipping, ensuring the impact of $g_r$ remains within $[-\gamma_r, \gamma_r]$.

Using an exponential decay function to model temporal decay was inspired by TLogic (Liu et al., 2022), which uses such a function to express temporal confidence. However, extensions are required to capture phenomena beyond TLogic's formulation. First, while TLogic applies a single fixed decay factor to all rules, we learn rule-specific parameters, enabling the representation of diverse temporal behaviours. Second, we introduce $\phi_r$ to control the asymptotic lower bound of $f_r$, allowing Count-TRuCoLa to capture rules that drop in confidence initially but keep a stable confidence afterwards.

Our approach learns a confidence function for each rule by optimizing the parameters $(\alpha_r, \lambda_r, \phi_r, \rho_r, \kappa_r, \gamma_r)$ aiming to minimize the sum of squared errors between the predicted confidence $conf_r(\Delta)$ (Equation 11) and the observed confidence values from the examples $E_r$. Recall that $E_r$ for a rule $r$ consists of $N$ examples $(\Delta_i, y_i)$, where $i = 1, \dots, N$. Here, $y_i$ is the truth value indicating whether a body and head grounding appeared together ($true = 1$) or not ($false = 0$). For simplicity, we omit the subscript $_r$ in the following.

Following the approach of Meilicke et al. (2024) for confidence computation in static KGs (see Section 4), we adjust the observed confidence values by adding a constant $\mathcal{P}$ to the denominator. Here, $\mathcal{P}$ is a hyperparameter. Unlike the static setting, we apply this transformation separately for each $\min(\Delta)$. We denote this transformation as $s$, yielding the modified observed confidence value $\tilde{y}_i = s(\Delta_i, y_i)$. Details of this computation are provided in Appendix A.2.

To learn the confidence function, we minimize the sum of squared errors (SSE) between the predictions $f + g$ and the transformed observed confidence values:

$$\underset{\alpha,\lambda,\phi,\rho,\kappa,\gamma}{\arg\min} \sum_{(\Delta_i, y_i) \in E} \left( f(\Delta_i) + g(\Delta_i) - s(\Delta_i, y_i) \right)^2. \tag{14}$$

We implement this in a two-step approach, where we first minimize the SSE to find values for $\alpha, \lambda, \phi$ assuming the contribution of $g$ is zero, and then, fixing the parameters of $f$ obtained in the first step, estimate $\rho, \kappa, \gamma$. This two-step approach represents a practical compromise. While it does not guarantee a globally optimal solution, in practice it significantly reduces training time without noticeably impacting prediction accuracy.

### 4.3 AGGREGATION

If CountTRuCoLa predicts a candidate $c$, it is typically predicted by several rules $r_0, ..., r_n$ with different confidence scores $s_0, ..., s_n$. We show an example in Section 6, Figure 2. Combining these confidence scores is known as rule aggregation problem (Betz et al., 2024). A common strategy for rule aggregation is the *noisy-or* strategy, which computes $1 - \prod_{i=0}^{n}(1 - s_i)$. Because noisy-or assumes independence among rules, we adopt the version proposed by Betz et al. (2024) that applies noisy-or to the top $\mathcal{H}$ confidences (e.g., $\mathcal{H} = 5$). We treat $\mathcal{H}$ as a hyperparameter. Additionally, we introduce a decay factor $\mathcal{D} \in [0, 1]$ as hyperparameter. Let $s_0, ..., s_n$ be sorted in descending order; we apply a modified noisy-or where each $s_i$ is replaced by $s_i \cdot \mathcal{D}^i$, giving less weight to lower-ranked scores. Appendix A.6 summarizes all hyperparameters.

### 4.4 INTERPRETABILITY

In this paper, we use the term interpretability in line with Molnar (2020), Biran & Cotton (2017) and Miller (2019), who describe it as "the degree to which a human can understand the cause of a decision" made by a model. For CountTRuCoLa, this means that each prediction can be directly linked to the specific rules that fired, their confidence scores, and the particular observations that support them. Concretely, CountTRuCoLa is interpretable due to

- Single-body rules: Each rule contains exactly one body atom. Thus, each rule's contribution depends on a single temporal triple and its observed occurrences over time (Section 4.1).
- Interpretable confidence function: Every rule's confidence is computed by a confidence function. The confidence function consists of a modified exponential decay combined with a clipped linear term, and is parametrized by six parameters who each have a specific and interpretable role (Section 4.2). These confidence values directly correspond to the rule scores used in prediction.
- Rule-based predictions: Each prediction is obtained by combining the scores of a fixed set of rules through a simple, transparent aggregation formula (Subsection 4.3). We further show that the aggregated rule confidence values correlate with empirical accuracies (Appendix A.8.1).
- Direct traceability to data: For any prediction, we can identify the exact triples and their temporal occurrences that activated each rule and contributed to the output.

In summary, the entire computation, from input, to rule firing, confidence values, to the final prediction, can be inspected directly, making CountTRuCoLa 's prediction process transparent and inherently interpretable, requiring no post-hoc auxiliary explanation method. The explanatory structure is identical to the computation performed by the model.

## 5 EXPERIMENTS

### 5.1 EXPERIMENTAL SETUP

For our experiments[1], we use the datasets from TGB 2.0 `smallpedia`, `polecat`, `icews`, `wikidata` (Gastinger et al., 2024a), and the datasets ICEWS14/18, GDELT, YAGO, WIKI, in the versions as used by Li et al. (2021b) and Gastinger et al. (2023). More information on the datasets including a table with dataset statistics can be found in Appendix A.4.

---

[1] `https://anonymous.4open.science/r/counttrucola_submission-A80B` contains anonymized code to reproduce our experiments.

Table 1: Model comparison on nine benchmark datasets. OOM means Out Of Memory (40 GB GPU or 1056 GB RAM), OOT means Out Of Time (7 days), - means that no results have been reported. Best results are shown in bold and underlined font, second-best bold, third-best underlined.

| | GDELT | | YAGO | | WIKI | | ICEWS14 | | ICEWS18 | | smallp. | | polecat | | icews | | wikidata | |
|---|---|---|---|---|---|---|---|---|---|---|---|---|---|---|---|---|---|---|
| | MRR | H10 | MRR | H10 | MRR | H10 | MRR | H10 | MRR | H10 | MRR | H10 | MRR | H10 | MRR | H10 | MRR | H10 |
| TRKG | 21.5 | 37.3 | 71.5 | 79.2 | 73.4 | 76.2 | 27.3 | 50.8 | 16.7 | 35.4 | - | - | - | - | - | - | - | - |
| xERTE | 18.9 | 32.0 | 87.3 | 91.2 | 74.5 | 80.1 | 40.9 | 57.1 | 29.2 | 46.3 | - | - | - | - | - | - | - | - |
| TANGO | 19.2 | 32.8 | 62.4 | 67.8 | 50.1 | 52.8 | 36.8 | 55.1 | 28.4 | 46.3 | - | - | - | - | - | - | - | - |
| Timetraveler | 20.2 | 31.2 | 87.7 | 91.2 | 78.7 | 83.1 | 40.8 | 57.6 | 29.1 | 43.9 | - | - | - | - | - | - | - | - |
| TiRGN | 21.7 | 37.6 | 88.0 | 92.9 | 81.7 | 87.1 | 44.0 | 63.8 | 33.7 | 54.2 | - | - | - | - | - | - | - | - |
| CognTKE | OOM | OOM | 90.6 | 93.2 | 83.2 | 87.3 | 46.1 | 64.5 | 35.2 | 54.7 | 53.4 | 70.1 | 28.7 | 45.5 | OOM | OOM | OOM | OOM |
| TLogic | 19.8 | 35.6 | 76.5 | 79.2 | 82.3 | 87.0 | 42.5 | 60.3 | 29.6 | 48.1 | 59.5 | 70.7 | 22.8 | 37.8 | 18.6 | 30.1 | OOT | OOT |
| RE-GCN | 19.8 | 33.9 | 82.2 | 88.5 | 78.7 | 84.7 | 42.1 | 62.7 | 32.6 | 52.6 | 59.4 | 68.7 | 17.5 | 29.2 | 18.2 | 33.1 | OOM | OOM |
| CEN | 20.4 | 35.0 | 82.7 | 89.4 | 79.3 | 84.9 | 41.8 | 60.9 | 31.5 | 50.7 | 61.2 | 70.5 | 18.4 | 32.3 | 18.7 | 33.4 | OOM | OOM |
| EdgeBank$_{tw}$ | 1.9 | 3.5 | 61.7 | 61.7 | 58.5 | 84.4 | 13.5 | 34.2 | 7.2 | 17.9 | 35.3 | 56.6 | 5.6 | 11.9 | 2.0 | 5.8 | 53.5 | 59.6 |
| Rec.B $(\psi_\triangle \xi)$ | 24.5 | 39.8 | 90.9 | 93.0 | 81.4 | 87.1 | 37.4 | 51.5 | 28.7 | 43.6 | 60.5 | 71.6 | 19.8 | 31.7 | 21.1 | 32.4 | OOT | OOT |
| CountTRuCoLa | 23.8 | 40.3 | 90.9 | 93.2 | 82.7 | 86.6 | 45.0 | 62.0 | 32.8 | 51.0 | 64.4 | 71.7 | 25.6 | 40.8 | 21.4 | 32.1 | 60.9 | 62.8 |

We follow the evaluation protocol from Gastinger et al. (2023), and use the TGB 2.0 evaluation framework (Gastinger et al., 2024a). We report time-aware filtered mean reciprocal rank (MRR) and Hits@10. All experiments are conducted for single-step prediction.

We compare our model to 9 methods of the 23 methods described in Section 2, as well as the two heuristic baselines EdgeBank (Poursafaei et al., 2022) and the Recurrency Baseline (Rec.B) (Gastinger et al., 2024b). Unless otherwise stated, we report results for these 11 methods based on the evaluations in Gastinger et al. (2024a) (for the TGB 2.0 datasets) and Gastinger et al. (2023) (for all other datasets). For TiRGN, we use the reported results and performed a sanity check of the released code. For CognTKE, we likewise use the reported results. Since CognTKE achieves the best performance among related methods, we additionally ran the authors' code on datasets not included in their study (GDELT, YAGO, and all TGB 2.0 datasets). We excluded 15 prior methods from comparison for reasons including unavailable code, incompatible evaluation protocols, or missing reproducibility information; detailed justifications are in Appendix A.5.1.

We tested CountTRuCoLa on a SLURM-managed CPU cluster with AlmaLinux 9.5 operating system. Nodes featured Intel Xeon (E5-2640 v2/v3/v4, Silver 4114) and AMD EPYC (7413, 7713P, 9474F) processors, with up to 96 cores and 1.5 TB RAM per node. However, parallelization was restricted to a maximum of 20 concurrent threads and we limited available RAM to 500 GB as the experiments were conducted in a shared environment. We did not use GPUs for these experiments. CognTKE experiments were run on a NVIDIA RTX A6000 GPU with 48 GB VRAM.

## 5.2 RESULTS

Table 1 reports test scores on all nine datasets[2]. CountTRuCoLa achieves the highest MRR on four datasets and the second or third highest on the remaining, indicating competitive performance. While Hits@10 performance is slightly lower, this is likely because hyperparameter tuning was performed based on MRR. CountTRuCoLa outperforms the Recurrency Baseline on 7 out of 9 datasets with substantial improvements on ICEWS14/18, smallpedia, and polecat, suggesting that rules beyond recurrency, combined with learned confidence functions, improve performance.

Unlike several existing approaches, which fail due to out-of-time (OOT) or out-of-memory (OOM) issues on large-scale graphs, CountTRuCoLa scales to all datasets without such failures. A runtime study (Appendix A.8.4) shows that CountTRuCoLa attains low runtimes compared to related work. This indicates that a lightweight, interpretable model can achieve predictive accuracy competitive with, and in some cases surpassing, more complex neural models while maintaining scalability.

**Comparison to CognTKE** CognTKE outperforms CountTRuCoLa on four datasets. One likely reason is CognTKE's local multi-hop reasoning, which propagates information across several hops of neighbors, and its temporal relation attention layers, which model reasoning paths of up to four hops (Chen et al., 2025). Through attention-based message passing, CognTKE adaptively combines multiple paths and features. These capabilities, however, come at the cost of higher total runtime and reduced transparency. As shown in Appendix A.8.4, CountTRuCoLa is over five times faster,

---

[2]We also report Hits@1 and Hits@3 performance in Appendix A.8.5.

even though it was run solely on CPU while CognTKE was executed on GPU. In addition, running CognTKE resulted in memory errors on the three largest datasets. Moreover, although CognTKE highlights relational subgraphs and attention-weighted paths, key steps remain black-box operations due to neural components, and lack the full transparency of our rule-based approach.

**A note on GDELT**  On GDELT, CountTRuCoLa underperforms the Recurrency Baseline despite outperforming all other models. GDELT, based on a platform that sources global news events in 15 minute intervals, features two temporal patterns: (i) short-term repetition driven by media coverage, and (ii) long-term dependencies, such as a country criticizing another in response to a military action. The difficulty of disentangling these temporal patterns likely explains the relative performance gap.

**Ablation Studies**  Table 2 shows ablation studies on ICEWS14 and WIKI. First, we analyze the contributions of different rule types, see part (a) of Table 2. For ICEWS14, $xy$-rules yield the highest MRR, which is expected since they form the core of our approach. Combining all rule types (row *all*) leads to the highest scores. However, the MRR scores do not simply add up due to overlapping predictions among rule types. We further evaluate the importance of each rule type by removing them individually. This leads to a performance decrease, with the highest drop observed when $xy$-rules are excluded. This highlights the benefit of incorporating multiple rule types. For WIKI, the ablation study reveals a different behavior. Here, using only recurrency rules already achieves the same results as combining all rule types. This suggests two things: First, capturing only very simple dependencies leads to strong predictive performance on this dataset; and second, CountTRuCoLa is robust against potential noise introduced by additional rules.

Table 2: Test scores for all rule types as introduced in Section 3, and for different scoring functions.

| | | WIKI | | ICEWS14 | |
|---|---|---|---|---|---|
| | | MMR | H10 | MRR | H10 |
| (a) Rule Types | rec-rules | 82.6 | 86.3 | 35.6 | 47.3 |
| | $xy$-rules (incl. rec.) | 82.5 | 86.3 | 42.8 | 60.4 |
| | $c$-rules | 29.6 | 30.0 | 25.2 | 34.6 |
| | $z$-rules | 14.1 | 24.9 | 14.1 | 27.9 |
| | $f$-rules | 67.5 | 82.8 | 34.0 | 46.5 |
| | all - $z$-rules | 82.6 | 86.4 | 44.7 | 61.6 |
| | all - $f$-rules | 82.6 | 86.5 | 43.5 | 61.3 |
| | all - $c$-rules | 82.6 | 86.5 | 44.8 | 61.7 |
| | all - ($xy$-rules (incl. rec.)) | 68.5 | 83.2 | 39.0 | 53.1 |
| | all | 82.6 | 86.6 | 45.0 | 62.0 |
| (b) Conf. Functions | static | 68.0 | 82.7 | 39.9 | 58.5 |
| | $g_r$ | 68.6 | 83.2 | 43.4 | 59.5 |
| | $f_r$ | 81.1 | 86.5 | 44.7 | 61.8 |
| | $f_r + g_r$ | 82.6 | 86.6 | 45.0 | 62.0 |
| (c) Params | fix $(\alpha, \lambda, \phi, \rho, \kappa, \gamma)$ | 77.9 | 86.6 | 38.0 | 56.8 |
| | fix $(\lambda, \phi, \rho, \kappa, \gamma)$ | 80.6 | 86.6 | 42.8 | 59.4 |
| | learn all | 82.6 | 86.6 | 45.0 | 62.0 |

Second, we analyze the benefit of our introduced scoring functions, see part (b) of Table 2, comparing four variants: (i) static confidence; (ii) using only $f$ (Equation 12), setting $g$ to zero, i.e., predicting from the most recent body grounding; (iii) using only $g$ (Equation 13), setting $f$ to zero, i.e., predicting based on frequency; (iv) the default setting combining $f$ and $g$ (Equation 11). For both datasets, combining $f$ and $g$ leads to considerable improvements. Each component individually also improves performance compared to static confidence. Incorporating frequency of body groundings further improves predictive capability compared to relying solely on recency.

Third, we evaluate whether learning the confidence-function parameters is necessary, see part (c) of Table 2. In variant (i), parameters $(\alpha, \lambda, \phi, \rho, \kappa, \gamma)$ are fixed to $(0.5, 0.1, 0, 0.01, 0, 1)$. In variant (ii), only $\alpha$ is adapted to the static confidence of each rule, while the remaining parameters are fixed. These settings result in an exponential curve with a mild decay for $f_r$ and a line with small slope for $g_r$. Variant (iii) corresponds to the full CountTRuCoLa approach. In (i), all rules share the same confidence curve. In (ii), only the starting value differs across rules. Both fixed-parameter settings reduce performance. This shows that learning rule-specific parameters is important, as different rules require different confidence-function shapes.

## 6 EXPLANATIONS

In addition to the forecasting model, CountTRuCoLa provides a tool that generates, for given queries, a webpage displaying candidate rankings and the rules contributing to each prediction. This tool is straightforward because, as described in Section 4.4, CountTRuCoLa is inherently interpretable. Each prediction can be traced back to the specific rules, their confidence scores, and the particular observations that contributed to it. The tool simply visualizes these steps.

```
Alexis_Tsipras Consult ? 334 Ground Truth Nodes: Evangelos_Venizelos

 1. Evangelos_Venizelos 0.22886
    1. 0.16944 (0.171+-0.001) Consult(X,Y,T) <= inv_Express_intent_to_meet_or_negotiate(X,Y,U) 1x  4  PLOT
    2. 0.03571 (0.036+0.0) Consult(X,Y,T) <= Criticize_or_denounce(X,Y,U) 0x  243  PLOT
    3. 0.03441 (0.034+0.0) Consult(X,Y,T) <= inv_Criticize_or_denounce(X,Y,U) 0x  78  PLOT
    4. 0.02831 (0.028+0.0) Consult(X,Y,T) <= Make_statement(X,Y,U) 0x  199  PLOT
    5. 0.02009 (0.02+0.0) Consult(X,Y,T) <= inv_Accuse(X,Y,U) 0x  78  PLOT
    6. 0.00038 (0.0+0) Z-rule: Consult(X, Evangelos_Venizelos, T) -1x  -1  PLOT
 2. New_Democracy 0.18268
    1. 0.09091 (0.091+0) F-rule: Consult(Alexis_T.,New_Democracy,T) <= ex.Consult(Alexis_T.,?,T) -1x  -1  PLOT
    2. 0.0406 (0.041+0.0) Consult(X,Y,T) <= inv_Make_an_appeal_or_request(X,Y,U) 0x  325  PLOT
    3. ...
 3. Antonis_Samaras 0.16653
    ...
```

Figure 2: Explanation for the query *(Alexis_T., Consult, ?, 334)* with ground truth *Evangelos_V.*

The tool supports several use cases: First, it assists model development by uncovering flaws and open problems. Second, it can enable comparison with other models, thereby identifying differences in predictions and the data patterns underlying them. And third, it provides explanations in real-world applications, showing reasons behind CountTRuCoLa's predictions and highlighting temporal patterns in a TKG. We illustrate the second use case with an example comparison between CountTRuCoLa and RE-GCN.

Figure 2 illustrates an explanation for the query *(Alexis_T., Consult, ?, 334)* with ground truth *Evangelos_V.* (ICEWS14). In this case, CountTRuCoLa ranks the correct entity at rank 1, whereas RE-GCN ranks it at position 8. The tool reveals that the top candidate's score ($\approx 0.22$) is largely due to a single rule, which states that individuals who expressed intent to meet earlier are likely to consult later. This rule has a confidence of 0.17 (recency score 0.171, frequency score $-0.001$), and was triggered by a triple occurring once, four timestamps prior, with a confidence function depicted in Figure 1. By clicking on the PLOT link, this figure is shown to users. Further, the top candidate's score it substantially higher than subsequent candidates. This explanation suggests a possible limitation of RE-GCN, that it may struggle to capture longer temporal dependencies (here, four timesteps) due to its restricted window size (three timesteps). Further predictions could be analyzed to examine whether this pattern occurs frequently or to uncover additional reasons for missed predictions.

# 7 DISCUSSION

In this work, we introduced a simple and scalable rule-based approach for TKG forecasting that is fully interpretable. Each prediction can be traced back to the regularity and the specific observation that triggered it. Despite its simplicity, our method achieves competitive performance across diverse datasets, outperforming most existing approaches and even reaching state-of-the-art results on some datasets.

The strong performance of our simple rule-based approach suggests that, at least for existing datasets, much of what is gained from complexity is in fact explainable by simple dependencies. Complex end-to-end architectures are not required to achieve competitive predictive results on the standard datasets. This implies that either these datasets do not contain richer patterns that demand complex modeling, or that most current models are not effectively capturing such patterns.

Beyond strong empirical results, our approach provides a transparent view into why predictions are made, allowing researchers to trace outcomes back to concrete patterns in the data. This not only makes CountTRuCoLa suitable for forecasting itself, but also offers a tool for understanding the behavior of existing methods and the properties of datasets.

Overall, our approach provides a practical and interpretable forecasting method, while contributing to better transparency, understanding, and future development of both rule-based and neural TKG forecasting systems, as well as their application to real-world use cases.

## REPRODUCIBILITY STATEMENT

We provide the full code in an anonymous repository[3], including scripts for running all steps discussed in the paper and for evaluation. All datasets used are publicly available, our code prompts the user to download them automatically. The repository includes a detailed README, configuration files, and a requirements file to install dependencies. We specify the experimental setup and exact hardware (model, CPU, and memory limits) in Section 5.1 and Appendix A.8.8. We report all hyperparameters and hyperparameter ranges in Appendix A.6. Although we did not fix random seeds, we show that test results vary only minimally across five independent runs (Appendix A.8.6) and we also report chi-squared significance tests in Appendix A.8.7. Upon acceptance, we will release the repository publicly on GitHub.

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

## A  APPENDIX

### A.1  ADDITIONAL INFORMATION ON COLLECTING EXAMPLES

The following pseudocode illustrates the procedure for collecting the example for a given $xy$-rule $r = h(x, y, t^*) \leftarrow b(x, y, t)$ as described in Section 4.

---
**Algorithm 1** Collecting Examples

---
1: **Input:** TKG $G$, $xy$-rule $r = h(x, y, t^*) \leftarrow b(x, y, t)$
2: **Output:** examples set $E_r$
3: $E_r = \emptyset$
4: **for** $t \in T(G)$ **do**
5:     **for** $c, d \in C(G)$ **do**
6:         **if** $\exists z \; h(c, z, t) \in G$ **then**
7:             $\Delta = \Delta_{\theta_{x/c, y/d}}^{r, G^t, t}$
8:         **if** $\Delta \neq \emptyset$ **then**
9:             **if** $h(c, d, t) \in G$ **then** add ($\Delta$, 1) to $E_r$
10:            **else** add ($\Delta$, 0) to $E_r$

---

Note that in Line 7, we collect only distances within a time window $\mathcal{W}$, where $\mathcal{W}$ is a hyperparameter.

### A.2  ADDITIONAL INFORMATION ON LEARNING THE CONFIDENCE FUNCTIONS

As described in Section 4, we transform the observed confidence value by introducing a scaling factor that depends on the smallest time distance. This transformation is controlled by a hyperparameter $\mathcal{P}$, which is added to the denominator when computing the confidence.

Below, we provide a detailed description of this transformation: For each group of examples in $E_r$ sharing the same smallest time distance $\min(\Delta_i)$, we compute a scaling factor $k(\min(\Delta_i))$ and use it to adjust the observed confidence value $y_i$. The transformation is defined as:

$$\tilde{y} = s(\Delta, y) = k(\min(\Delta)) \cdot y, \tag{15}$$

The following pseudo-code (Algorithm 2) outlines the computation of $k$:

---
**Algorithm 2** Computing the scaling factor $k$

---
1: **Input:** Example Set $E$ for rule $r$, Parameter $\mathcal{P}$
2: **Output:** Mapping $k$ from minimal distance $d$ to scaling factor
3: Initialize empty set $X = \emptyset$ and empty maps $p, a, k$
4: **for** each $(\Delta_i, y_i)$ in $E$ **do**
5:     $d = \min(\Delta_i)$                  // minimal distance
6:     **if** $d \notin X$ **then**
7:         Add $d$ to $X$, $p(d) = 0$, $a(d) = 0$
8:     $p(d) = p(d) + y_i$           // count positive examples for $d$
9:     $a(d) = a(d) + 1$              // count all examples for $d$
10: **for** each $d$ in $X$ **do**
11:     $k(d) = \frac{a(d)}{a(d) + \mathcal{P}}$         // compute scaling factor for $d$
    **return** $k$

---

This approach ensures that the observed confidence values are adjusted according to the number of examples for each smallest time distance, controlled by the hyperparameter $\mathcal{P}$.

### A.3  SPECIAL REMARK ON $c$-RULES

In this section we are providing a more detailed consideration of $c$-rules with the help of an example. Consider a dataset in which a regularity is observed: if a person eats pizza at timestamp $t$, they will

drink espresso at a subsequent timestamp. This pattern can be represented by the following $c$-rule, where $e$ denotes eat and $d$ denotes drink:

$$d(x, espresso, t^*) \leftarrow e(x, pizza, t) \land t^* > t \tag{16}$$

This $c$-rule can be directly applied, to answer a query such as $drinks(x, ?, t^*)$, to determine what person $x$, who just ate pizza, will drink.

The mentioned regularity can also be used to answer a different type of query: Who will drink an espresso at timestamp $t^*$, expressed as $drinks(?, espresso, t^*)$, or equivalently, using inverse relations, as $drinks^{-1}(espresso, x, t^*)$.

Intuitively this could be expressed by such a rule:

$$d^{-1}(espresso, x, t^*) \leftarrow e^{-1}(pizza, x, t) \land t^* > t \tag{17}$$

However, our language bias does not support rules where the constant appears in the subject position, meaning Rule (17) is not covered by our language bias.

Luckily, there is an inverse relation for each relation. This means that we can express the regularity as

$$d(x, espresso, t^*) \leftarrow e(x, pizza, t) \land t^* > t \tag{18}$$

Rule (18) is covered by our language bias.

The fact that Rule (18) looks the same as Rule (16) is because so far we have ignored an important detail: We have omitted the additional body atom, which ensures that the query is asked only if there is at least one correct answer. It makes a difference whether such a rule should answer queries such as $drinks(x, ?, t^*)$, that we call forward queries, or queries such as $drinks(?, espresso, t^*)$, which we call backward queries. For the forward query the additional body atom is $drinks(x, z, t^*)$, for the backward query $drinks(z, espresso, t^*)$,

This body atom does not only make a difference in the formal writing of the rule, but also when collecting the examples $E_r$ and learning the parameters for the rule. For both rules, we need to find cases where people drink espresso after eating pizza as positive examples. However the negative examples differ: For the first query, $drinks(x, ?, t^*)$, people who drink tea after eating pizza would be negative examples. For the second case, $drinks(?, espresso, t^*)$, the negative example would be if one person eats pizza and in the next timestamp, another person drinks espresso.

Thus, to support any possible application of a $c$-rule, we have to compute two variants of the same $c$-rule that differ only with respect to the additional atom:

$$h(x, d, t^*) \leftarrow b(x, d', t) \land \exists z \, h(x, z, t^*) \land t^* > t \tag{19}$$

$$h(x, d, t^*) \leftarrow b(x, d', t) \land \exists z \, h(z, d, t^*) \land t^* > t \tag{20}$$

where Rule (19) belongs to the query such as $drinks(x, ?, t^*)$, and Rule (20) belongs to the query such as $drinks(?, espresso, t^*)$. We call Rule (19) the forward version of a $c$-rule ($c$-rule $F$), and Rule (20) the backward version of a $c$-rule ($c$-rule $B$).

Depending on the query that we have to answer, we use the appropriate variant of the relevant $c$-rules. In Line 6 in Algorithm 1 we have to use the additional atom of the forward variant to compute the example set for the first variant, or the additional atom of the backward variant to compute the example set for the second variant.

### A.4 ADDITIONAL INFORMATION ON DATASETS

We provide additional details on the datasets used in our experiments. For each dataset, we use the version provided by Li et al. (2021b) and Gastinger et al. (2023), or, if lowercase letters only, the datasets introduced by Gastinger et al. (2024a). An overview on the statistics of the datasets is in Table 3.

**ICEWS Datasets:** ICEWS14 (García-Durán et al., 2018), ICEWS18 (Jin et al., 2019), and icews (Gastinger et al., 2024a) are derived from the Integrated Crisis Early Warning System (ICEWS) (Boschee et al., 2015; Shilliday et al., 2012). These datasets span different periods

Table 3: Dataset statistics. # Quads refers to the number of quadruples without inverse quadruples.

| Dataset | smallpedia | polecat | icews | wikidata | ICEWS14 | ICEWS18 | GDELT | YAGO | WIKI |
|---|---|---|---|---|---|---|---|---|---|
| # Quads Train | 387,757 | 1,246,556 | 10,861,600 | 6,982,503 | 74,845 | 373,018 | 1,734,399 | 161,540 | 539,286 |
| # Quads Valid | 81,033 | 266,736 | 2,326,157 | 1,434,950 | 8,514 | 45,995 | 238,765 | 19,523 | 67,538 |
| # Quads Test | 81,586 | 266,318 | 2,325,689 | 1,438,750 | 7,371 | 49,545 | 305,241 | 20,026 | 63,110 |
| # Nodes | 47,433 | 150,931 | 87,856 | 1,226,440 | 7,128 | 23,033 | 7,691 | 10,623 | 12,554 |
| # Relations | 283 | 16 | 391 | 596 | 230 | 256 | 240 | 10 | 24 |
| # Timestamps | 125 | 1,826 | 10,224 | 2,025 | 365 | 303 | 2,975 | 188 | 231 |
| Granularity | year | day | day | year | day | day | 15 min | year | year |

(2014, 2018, and 1995–2022) and contain event data on global political activities such as conflicts, protests, and diplomatic interactions. Events are categorized according to the CAMEO taxonomy (Gerner et al., 2002).

**GDELT:** The Global Database of Events, Language, and Tone (GDELT) (Leetaru & Schrodt, 2013) contains large-scale event data extracted from global news sources. It encompasses a wide range of political, societal, and cultural events across various countries and timeframes.

**polecat:** Based on the POLECAT (POLitical Event Classification, Attributes, and Types) dataset (Scarborough et al., 2023), this dataset records cooperative and hostile interactions between socio-political actors. POLECAT uses the PLOVER ontology (Halterman et al., 2023) and automated NLP pipelines to classify and extract time-stamped, geolocated events from multilingual news sources. The dataset used in this work covers the period from January 2018 to December 2022.

**YAGO and WIKI:** YAGO (Mahdisoltani et al., 2015) and WIKI Leblay & Chekol (2018) provide structured knowledge graph data with temporal relations. WIKI is extracted from Wikidata (Vrandečić & Krötzsch, 2014) and both datasets have been further processed by Jin et al. (2019) to represent temporal facts as quadruples. Events before 1786 (WIKI) and 1830 (YAGO) are excluded.

**smallpedia and wikidata:** These datasets are constructed from Wikidata (Vrandečić & Krötzsch, 2014) and processed by Gastinger et al. (2024a). smallpedia includes entities with IDs below 1 million, while wikidata extends the scope to entities with IDs up to 32 million. Both datasets contain event-based (point-in-time) and fact-based (duration) temporal relations between entities.

## A.5 DETAILS ON EXPERIMENTAL SETUP

### A.5.1 REASONS FOR EXCLUDING PRIOR METHODS FROM COMPARISON

As noted in Section 5.1, we exclude 15 prior methods from direct comparison due to various limitations. In this section, we provide detailed justifications for each exclusion.

INFER reports result on the "best" ranking protocol in the presence of ties. As a result, the evaluation always assigns the best possible rank to the ground-truth entity in the case of ties, rather than using an average or random tie-breaking strategy. This protocol is known to inflate evaluation metrics unfairly, since it does not reflect the true ranking uncertainty in the presence of score ties (Sun et al., 2020). Moreover, the provided code cannot be executed as it lacks the necessary specifications for pretraining the required Complex model. L2TKG, TPAR, Logenet, TempValid, ALREIR and TECHS do not provide code to reproduce results. CENET, RETIA, and CluSTER do not report results in time-aware filter setting. CyGNet and RE-Net run only in multi-step setting, not in single-step setting. TR-Rules uses different dataset versions for ICEWS14 and does not report results on WIKI, YAGO, GDELT, or the TGB 2.0 datasets. GenTKG evaluates on different versions of GDELT and YAGO, does not report results on WIKI or the TGB 2.0 datasets, and does not provide MRR scores for any dataset. Finally, we do not compare to zrLLM because it uses a different evaluation setup, focusing on zero-shot relations with different datasets and on predicting previously unseen relations.

### A.5.2 NEGATIVE SAMPLES

Following the TGB 2.0 evaluation framework, we use the provided negative samples for the large dataset tkgl-wikidata. Specifically, TGB 2.0 includes 1,000 negative samples per query, sam-

Table 4: Hyperparameter ranges. We allow fewer values for larger datasets to reduce computation costs.

| | small
WIKI, ICEWS14, YAGO, smallpedia | medium
ICEWS18 | large
polecat, wikidata, GDELT | very large
icews |
|---|---|---|---|---|
| $\mathcal{P}$ (RULE_UNSEEN_NEGATIVES) | $\{0, 1, 2, 3, 5, 10, 20, 30, 100\}$ | $\{1, 5, 10, 30, 100\}$ | $\{1, 10, 30, 100\}$ | $\{1, 10, 30, 100\}$ |
| $\mathcal{P}_{f\text{-rules}}$ (F_UNSEEN_NEGATIVES) | $\{0, 1, 5, 10, 20, 30\}$ | $\{0, 10, 30\}$ | $\{10\}$ | $\{10\}$ |
| $\mathcal{C}$ (RULE_TYPE_C) | $\{$True, False$\}$ | $\{$True, False$\}$ | $\{$True, False$\}$ | $\{$False$\}$ |
| $\mathcal{W}$ (LEARN_WINDOW_SIZE) | $\{50, 100, 150\}$ ICEWS14, $\{2, 3, 5, 10, 30, 50, 100\}$ WIKI, $\{2, 3, 5, 10, 30, 50, 100\}$ smallpedia, $\{2, 3, 5, 10, 30, 30, 50\}$ YAGO | $\{50, 100, 150\}$ | $\{50, 100, 150\}$ polecat, $\{2, 3, 5, 10, 30, 50\}$ wikidata, $\{10, 30, 50, 100, 150, 200\}$ GDELT | $\{50, 100\}$ |
| $\mathcal{M}$ (DATAPOINT_THRESHOLD_MULTI) | $\{0, 10, 50\}$ | $\{0, 50\}$ | $\{0, 50\}$ | $\{50\}$ |
| $\mathcal{Z}$ (Z_RULES_FACTOR) | $\{0, 0.1, 0.2, 0.3, 0.4, 0.5, 1.0\}$ | $\{0, 0.1, 0.5\}$ | $\{0, 0.1, 0.5\}$ | $\{0, 0.1, 0.5\}$ |
| $\mathcal{H}$ (NUM_TOP_RULES) | $\{5, 10, 50\}$ | $\{10, 50\}$ | $\{10\}$ | $\{10\}$ |
| $\mathcal{D}$ (AGGREGATION_DECAY) | $\{1, 0.9, 0.8, 0.7, 0.6, 0.5, 0.4\}$ | $\{1, 0.8, 0.6\}$ | $\{0.8\}$ | $\{0.8\}$ |

Table 5: Hyperparameter values for each dataset, selected based on the validation MRR. The column `default` represents default hyperparameters that we represent, when no hyperparameter tuning can or should be conducted.

| | GDELT | YAGO | WIKI | ICEWS14 | ICEWS18 | smallp. | polecat | icews | wikidata | default |
|---|---|---|---|---|---|---|---|---|---|---|
| $\mathcal{P}$ (RULE_UNSEEN_NEGATIVES) | 30 | 30 | 1 | 30 | 100 | 3 | 30 | 100 | 30 | 30 |
| $\mathcal{P}_{f\text{-rules}}$ (F_UNSEEN_NEGATIVES) | 10 | 30 | 10 | 10 | 10 | 30 | 10 | 10 | 10 | 10 |
| $\mathcal{C}$ (RULE_TYPE_C) | False | True | True | True | True | True | False | False | True | True |
| $\mathcal{W}$ (LEARN_WINDOW_SIZE) | 200 | 30 | 3 | 50 | 50 | 3 | 150 | 50 | 10 | 10 |
| $\mathcal{M}$ (DATAPOINT_THRESHOLD_MULTI) | 0 | 50 | 50 | 0 | 50 | 10 | 0 | 50 | 0 | 0 |
| $\mathcal{Z}$ (Z_RULES_FACTOR) | 0.1 | 0.1 | 0.1 | 0.1 | 0.1 | 0 | 0.1 | 0.1 | 0.1 | 0.1 |
| $\mathcal{H}$ (NUM_TOP_RULES) | 10 | 5 | 10 | 10 | 10 | 5 | 10 | 10 | 10 | 10 |
| $\mathcal{D}$ (AGGREGATION_DECAY) | 0.8 | 0.7 | 0.8 | 0.8 | 0.8 | 0.4 | 0.8 | 0.8 | 0.8 | 0.8 |

pled based on the query relation type. As a result, evaluation on `tkgl-wikidata` is performed by ranking the correct entity against these 1,000 candidates rather than against all entities. For all other datasets, we compute scores by ranking against the full set of entities.

## A.6 HYPERPARAMETERS

CountTRuCoLa comes with eight hyperparameters:

- $\mathcal{P}$: A constant added to the denominator when computing confidences for $xy$- and $c$-rules.

- $\mathcal{P}_{f\text{-rules}}$: The corresponding constant used for $f$-rules.

- $\mathcal{C}$: A boolean flag indicating whether $c$-rules are used for a given dataset.

- $\mathcal{W}$: The window size defining the length of the time window for the examples $E$, i.e., the maximum value in $\Delta$.

- $\mathcal{M}$: A threshold defining the minimum number of data points required before learning the parameters of the frequency-based scoring function $g_r$. If a rule has fewer than $\mathcal{M}$ examples, all parameters of $g_r$ are set to zero to reduce noise.

- $\mathcal{Z}$: A scaling factor $\mathcal{Z} \in [0, 1]$ applied to the score predicted by the $z$-rules.

- $\mathcal{H}$ and $\mathcal{D}$: Two hyperparameters for rule aggregation. $\mathcal{H}$ specifies how many of the top-confidence rules to aggregate, and $\mathcal{D} \in [0, 1]$ is a decay factor used in our modified noisy-or model, where the $i$-th confidence score $s_i$ is weighted by $s_i \cdot \mathcal{D}^i$.

We perform grid search on the validation MRR to select the values of these hyperparameters for each dataset. For smaller datasets, we explore a broader range of values, while for larger datasets, we reduce the search space to limit runtime and energy consumption. Table 4 lists the tested ranges for each dataset. Note that the range for $\mathcal{W}$ varies depending on the total number of timesteps available in each dataset. Table 5 reports the hyperparameter values selected for each dataset.

Table 6: Test MRRs when selecting hyperparameters for each dataset (based on the validation MRR) vs. when using standard hyperparameter values.

|  | GDELT | YAGO | WIKI | ICEWS14 | ICEWS18 | smallp. | polecat | icews |
|---|---|---|---|---|---|---|---|---|
| selected per dataset | 0.238 | 0.909 | 0.826 | 0.450 | 0.328 | 0.644 | 0.256 | 0.214 |
| default hyperparams | 0.203 | 0.908 | 0.812 | 0.444 | 0.312 | 0.630 | 0.250 | 0.208 |

## A.7 HYPERPARAMETER SENSITIVITY

Figure 3 shows the effect of selected hyperparameters on the test MRR across six datasets. We analyze the impact of $\mathcal{H}$ (NUM_TOP_RULES), $\mathcal{D}$ (AGGREGATION_DECAY), and $\mathcal{W}$ (LEARN_WINDOW_SIZE). Note that the hyperparameters selected for CountTRuCoLa (Table 5) are based on validation performance and do not necessarily result in the highest test MRR.

**Learn Window Size $\mathcal{W}$:** Across all datasets, $\mathcal{W}$ has the largest effect on performance. For the Wikidata- and Yago-based datasets (WIKI, YAGO, smallpedia), the best results are achieved with a small window of 3, while the other datasets benefit from larger windows. This can likely be explained by two factors. First the Temporal granularity: Yearly datasets (WIKI, YAGO, smallpedia) cover multiple years even with small windows, whereas daily (ICEWS14, ICEWS18, polecat) or 15-min (GDELT) datasets require larger windows to capture sufficient history; Second, the Recurrency of facts: As observed by Gastinger et al. (2024b;a), WIKI, YAGO, and smallpedia exhibit high direct recurrency ($> 85\%$), i.e., most temporal triples repeat in the previous timestep. Leveraging only recent history is sufficient, while incorporating older facts may introduce noise.

**Number of Top Rules $\mathcal{H}$:** For most datasets, performance improves with larger $\mathcal{H}$, with no substantial improvements between $\mathcal{H} = 10$ and $\mathcal{H} = 50$. Wikidata- and Yago-based datasets show a slight performance decrease as $\mathcal{H}$ increases, though the effect is minor. For a more detailed analysis of the effect of rule aggregation we refer to Section A.8.1.

**Aggregation Decay $\mathcal{D}$:** The aggregation decay $\mathcal{D}$ controls the assumed correlation between rules. $\mathcal{D} = 0$ corresponds to max-aggregation strategy, i.e., equals $\mathcal{H} = 1$, assuming highly correlated rules. In contrast, $\mathcal{D} = 1$ corresponds to noisy-or top-$\mathcal{H}$ aggregation, assuming independence between rules. For most datasets, intermediate values ($0.5 \leq \mathcal{D} \leq 0.8$) yield the best performance, reflecting partial correlation between rules. This aligns with the intuition that some rules are correlated, but not all. Wikidata-based datasets achieve the highest MRR with small $\mathcal{D}$ and $\mathcal{H}$, likely because predictions are dominated by recurrent rules, and additional correlated rules provide limited benefit.

**Comparison to Default Hyperparameters** Table 6 compares test MRR when hyperparameters are selected based on validation performance (as reported in Appendix A.6 and Table 5) versus using default hyperparameter values, which are reported in the last column of Table 5 and were chosen based on intuition and general insights. The results indicate that hyperparameter selection generally improves performance. For example, the difference for GDELT ranges from 20.3 to 23.8, whereas other datasets, such as polecat and YAGO, are relatively stable, with differences of 0.6 and 0.1, respectively.

## A.8 ADDITIONAL RESULTS

### A.8.1 RULE AGGREGATION ANALYSIS

To study how aggregation affects the relationship between predicted confidence and empirical correctnes we analyze the correspondence between aggregated confidence scores and ground-truth correctness under different aggregation configurations. This analysis allows us to study how the aggregation hyperparameter $\mathcal{H}$ (NUM_TOP_RULES) and the decay hyperparameter $\mathcal{D}$ (AGGREGATION_DECAY) influence the aggregated confidence values and the extent to which correlated rules may inflate them.

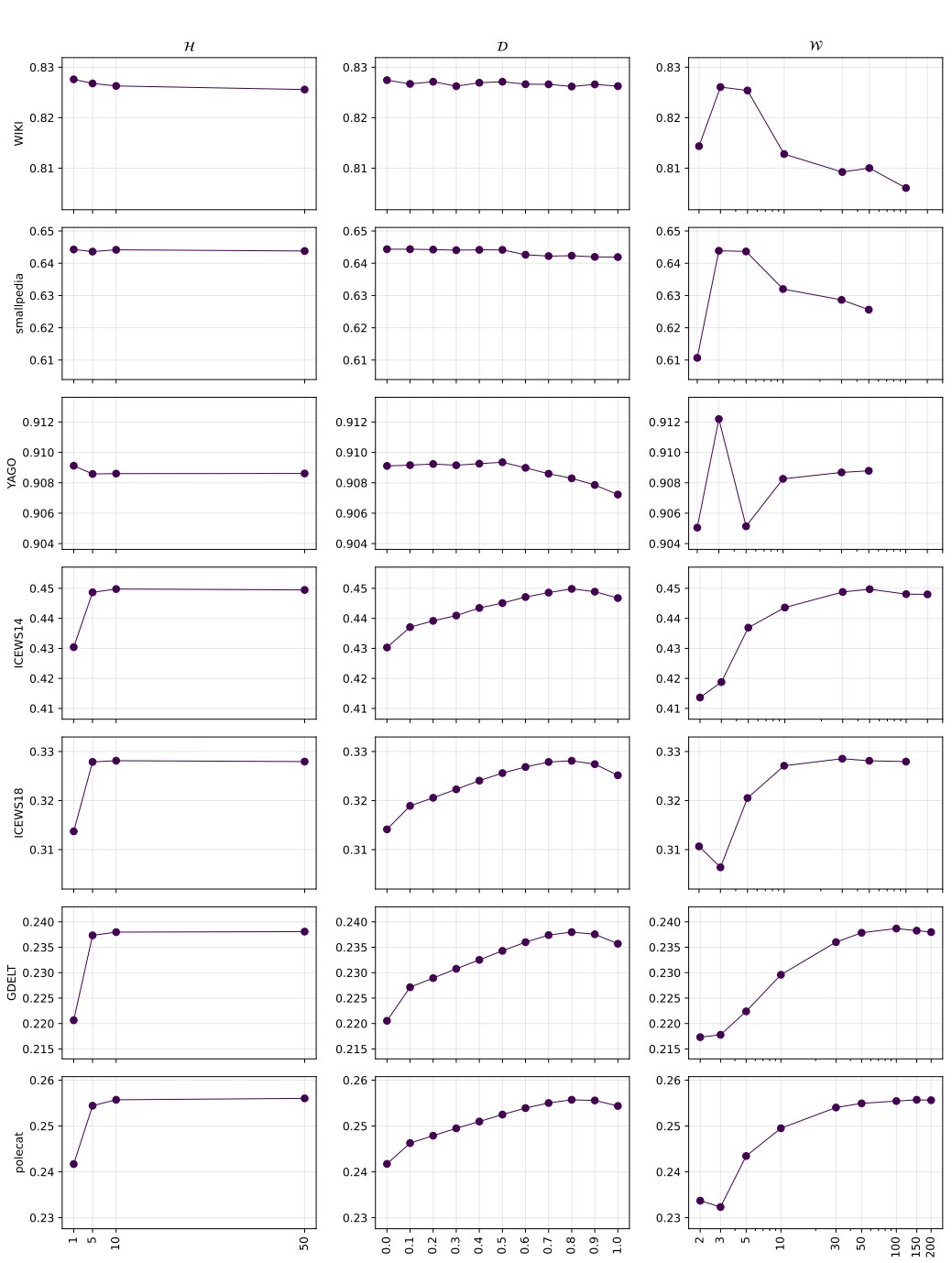

Figure 3: Effect of the hyperparameters $\mathcal{H}$ (NUM_TOP_RULES), $\mathcal{D}$ (AGGREGATION_DECAY), and $\mathcal{W}$ (LEARN_WINDOW_SIZE) on the test MRR. Columns correspond to hyperparameters (left to right), and rows correspond to datasets.

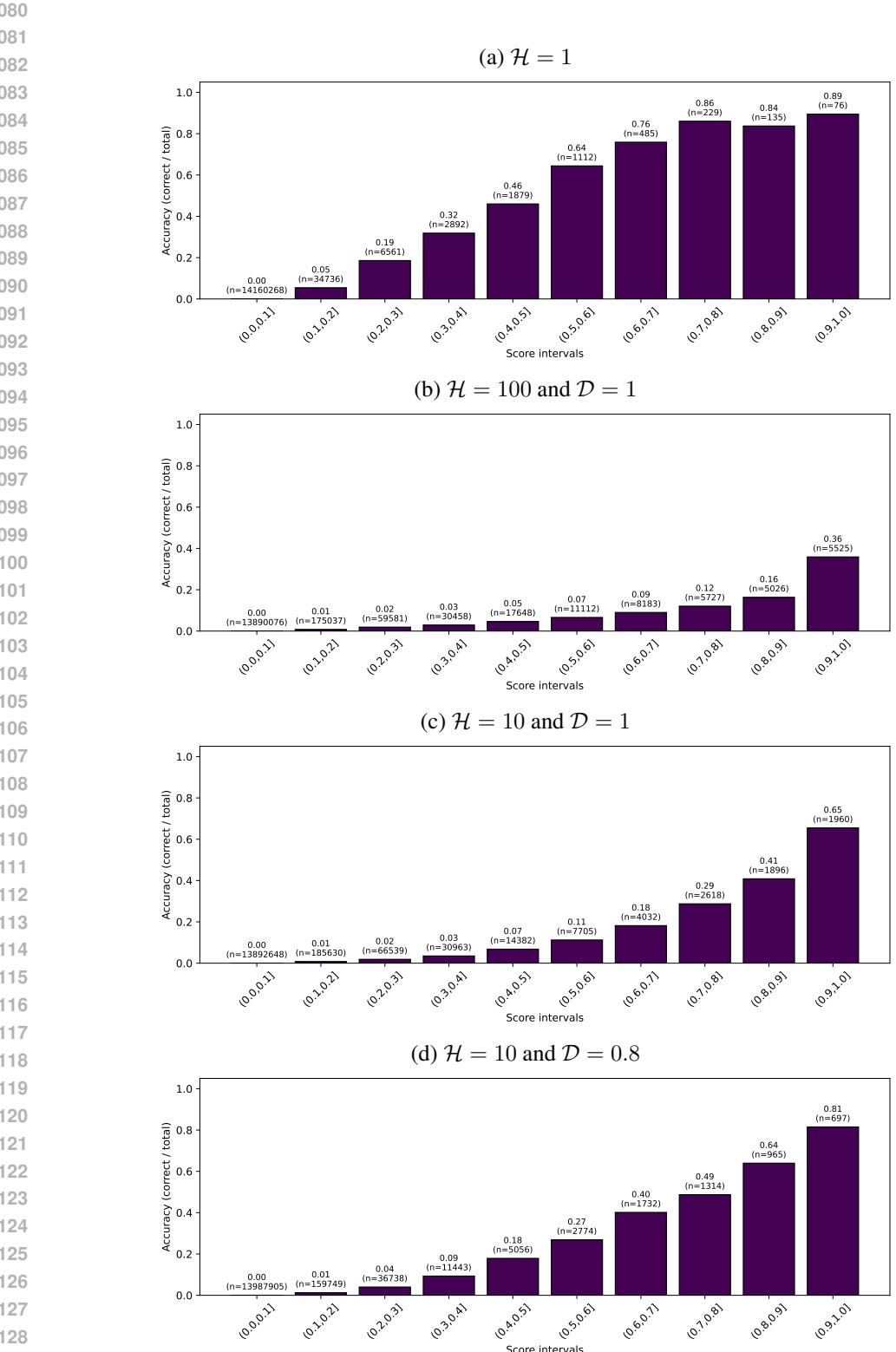

Figure 4: Analysis of the relationship between aggregated rule confidence and empirical correctness for ICEWS14. Candidates are grouped into ten confidence bins, and the fraction of correct predictions in each bin is reported for four aggregation settings, where $\mathcal{H}$ represents (NUM_TOP_RULES), and $\mathcal{D}$ (AGGREGATION_DECAY).

Table 7: Hits@1 and the mean score assigned to the top-scoring candidate for different $(\mathcal{H}, \mathcal{D})$ settings on `ICEWS14`.

| $\mathcal{H}$ | $\mathcal{D}$ | Hits1 | Mean Score |
|---|---|---|---|
| 1 | 1 | 32.9 | 27.1 |
| 100 | 1 | 34.0 | 56.6 |
| 10 | 1 | 35.6 | 51.3 |
| 10 | 0.8 | 35.9 | 42.2 |

**Analysis of Over- and Underestimation in Confidence Bins**   In this analysis, for each test query, we collected all predicted candidates together with their aggregated confidence scores and ground-truth labels (true or false). We then group the candidates in ten confidence bins $(i, i + 0.1]$ with $i$ in $0, 0.1, 0.2, \ldots, 0.9$ based on their assigned confidence score. For each bin, we computed the fraction of correct predictions.

The resulting curves indicate whether higher aggregated scores correspond to higher accuracy, thereby providing a diagnostic of whether CountTRuCoLa produces meaningful confidence values. We report results for four aggregation strategies: (a) max-aggregation ($\mathcal{H} = 1$), which uses only the highest-scoring rule and thus assumes strong dependence among rules; (b) top-100 noisy-or aggregation ($\mathcal{H} = 100$ and $\mathcal{D} = 1$), which combines the 100 highest-scoring rules under a stronger independence assumption; (c) top-10 noisy-or aggregation ($\mathcal{H} = 10$ and $\mathcal{D} = 1$), which combines the ten highest-scoring rules under a slightly weaker independence assumption; and (d) decayed noisy-or aggregation ($\mathcal{H} = 10$ and $\mathcal{D} = 0.8$) which reduces the influence of lower-ranked rules and thus aims to mitigate potential inflation from correlated rule sets.

Figure 4 reports the results for ICEWS14. The max-aggregation strategy (a) exhibits underestimation: Specifically, bins in the range 0.5–0.8 achieve higher empirical accuracies than their assigned confidence values, and relatively few candidates receive scores above 0.9. In contrast, the top-100 noisy-or strategy (b) shows substantial overestimation, with predicted scores exceeding accuracy across all bins and a substantially larger fraction of candidates receiving scores above 0.9. The top-10 noisy-or strategy (c) still overestimates confidence, but the inflation effect is noticeably reduced. Finally, the decayed noisy-or strategy (d) shows intermediate behavior. While a slight degree of overestimation remains, its predicted confidence values more closely match the empirical accuracies, and a monotonic relationship between confidence bins and accuracy is preserved.

**Analysis of Over- and Underestimation in Top-1 Predictions**   To complement the analysis above, we further examine the confidence assigned to the top-ranked predictions. Table 7 reports, for different $(\mathcal{H}, \mathcal{D})$ settings, the achieved Hits@1 and the mean score assigned to the top-scoring candidate on `ICEWS14`. Ideally, the assigned score should align with the achieved Hits@1. Systematically higher scores indicate overestimation, for example due to overcounting correlated rules, while systematically lower scores indicate underestimation.

The results reflect the expected behavior. Max-aggregation underestimates confidence: with $\mathcal{H} = 1$ and $\mathcal{D} = 1$, the model assigns a lower score than the observed Hits@1, reflecting overly strong dependence assumptions. In contrast, a large $\mathcal{H} = 100$ leads to clear overestimation, with the predicted score far exceeding the Hits@1. This demonstrates that noisy-or indeed overcounts correlated rules when many rules are aggregated. A moderate setting of $\mathcal{H} = 10$ partially alleviates this effect, though still produces overconfident estimates. Introducing a decay factor ($\mathcal{H} = 10$, $\mathcal{D} = 0.8$) results in a closer match between predicted scores (42.2) and Hits@1 (35.9), while also producing the highest Hits@1 among the tested configurations.

Together with the trends in Figure 4, these results provide empirical evidence that, while not being able to fully hinder them, the decay parameter mitigates overcounting effects.

### A.8.2   ANALYSIS OF PARAMETER SENSITIVITY TO LEARN WINDOW SIZE

Figures 5 and 6 illustrate the impact of the Learn Window Size $\mathcal{W}$ on the distribution of learned parameters across the datasets `ICEWS14`, `GDELT`, and `WIKI`.

Recall that the recency function $f$ is parameterized by $\lambda, \alpha, \phi$, which respectively control the exponential rate of decay, the starting value at $\Delta = 1$, and lower bound of the curve, while the frequency function $g$ is parameterized by $\rho, \kappa$, and $\gamma$, controlling its slope, shift, and value clipping. Each rule learns its own set of these parameters.

Figure 5 summarizes the effect of $\mathcal{W}$ on the learned recency curves, showing for each dataset the mean curve together with its 10th and 90th percentile variation bands for three representative window sizes. The figure also includes violin plots depicting, across rules, the distributions of $\lambda, \alpha$, and $\phi$ for different $\mathcal{W}$.

For the yearly dataset WIKI, the learned $\alpha$ values are consistently larger than in the other datasets across all window sizes, indicating that rules assign comparatively high confidence to events observed at $\Delta = 1$. The values of $\lambda$ are also generally higher, leading to steeper decay curves. This behavior stands in contrast to GDELT and ICEWS14, whose learned $\alpha$ values are substantially smaller, indicating lower confidence in general. Both of these datasets have much finer temporal granularity (15-minute and daily), so events are observed at a much higher density. In such settings, a single additional day or timestep carries less significance, and the model correspondingly learns shallower decay.

Despite differences in scale, GDELT and ICEWS14 exhibit similar trends as $\mathcal{W}$ increases. Larger windows tend to reduce both $\alpha$ and $\phi$, while the smallest $\lambda$ values appear at intermediate window sizes. The variance of $\lambda$ grows with $\mathcal{W}$, suggesting that broader learning windows allow the rules to specialize more strongly. Some rules learn sharply decaying recency curves, whereas others maintain relevance more gradually over longer time spans, leading to a diverse collection of temporal behaviors.

Figure 6 presents the corresponding analysis for the frequency function. For ICEWS14 and GDELT, increasing $\mathcal{W}$ leads to frequency curves that become both steeper and higher. An interpretation is that with more historical context available, CountTRuCoLa places greater weight on frequency and learns to differentiate reliable, frequently recurring patterns from noisy or isolated events. At small $\mathcal{W}$, by contrast, frequency becomes less informative, which is reflected in flatter curves. At the extreme case of ICEWS14 with $\mathcal{W} = 3$, some rules even learn significantly negative $\rho$, indicating situations in which additional occurrences reduce the score, whic is likely an effect of noise or event sparsity in very short windows. Such behavior disappears at larger windows and is less occurent for the other datasets. WIKI shows a different pattern. Its frequency curves remain comparatively flat for all window sizes, and the distributions of $\rho$ and $\kappa$ show smaller variation.

Taken together, these observations show that CountTRuCoLa does not converge to a single temporal pattern. Instead, it learns a diverse set of curves, with substantial variation in parameters such as $\lambda, \rho$, and $\kappa$ across rules. GDELT and ICEWS14 exhibit similar parameter behaviors across window sizes, whereas WIKI behaves fundamentally differently due to its yearly resolution and high direct recurrence.

Importantly, the results provide no evidence that recency or frequency parameters are transferable across datasets with different temporal granularities. Rather, the fitted parameters adapt strongly to the underlying time scale: coarser datasets learn sharper recency effects and flatter frequency responses, while finer-grained datasets support a wide spectrum of decay rates and frequency sensitivities. Thus, the choice of window size interacts closely with temporal resolution, and the learned parameters remain dataset-specific rather than universal.

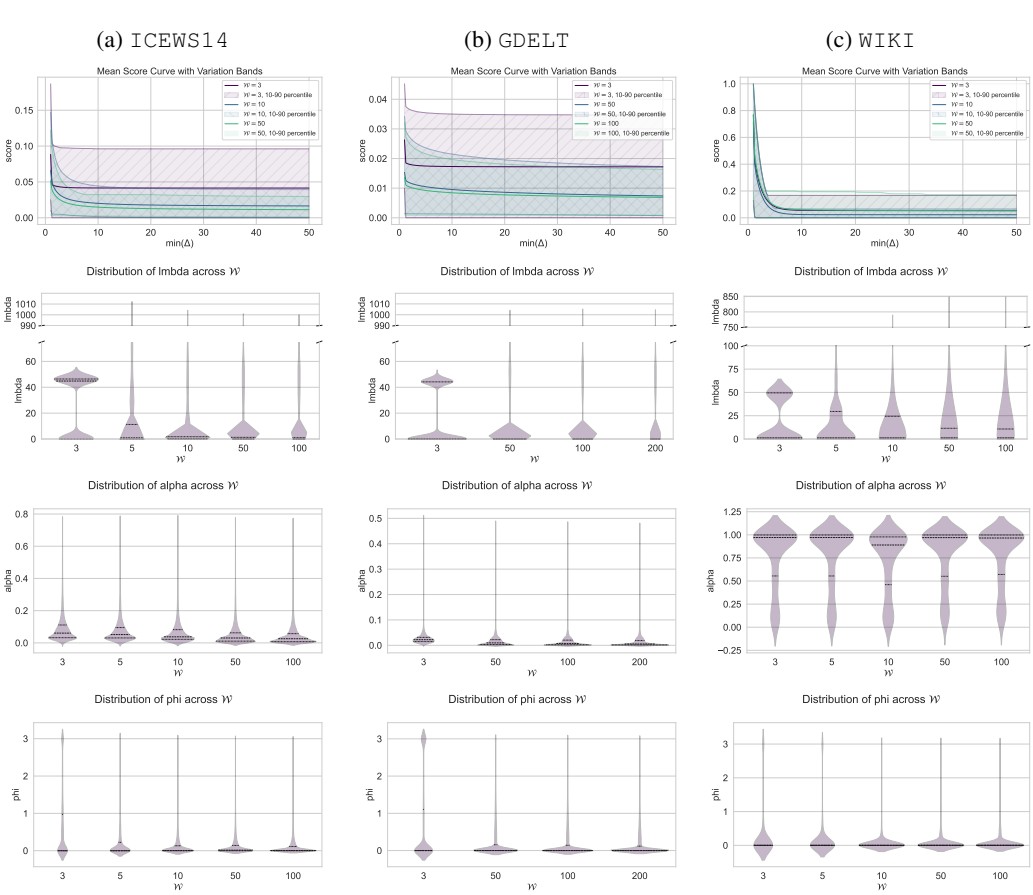

Figure 5: Recency function $f$ and the effect of Learn Window Size $\mathcal{W}$ on learned parameters for datasets ICEWS14, GDELT, and WIKI (left to right). Note that the y-axis varies across datasets.

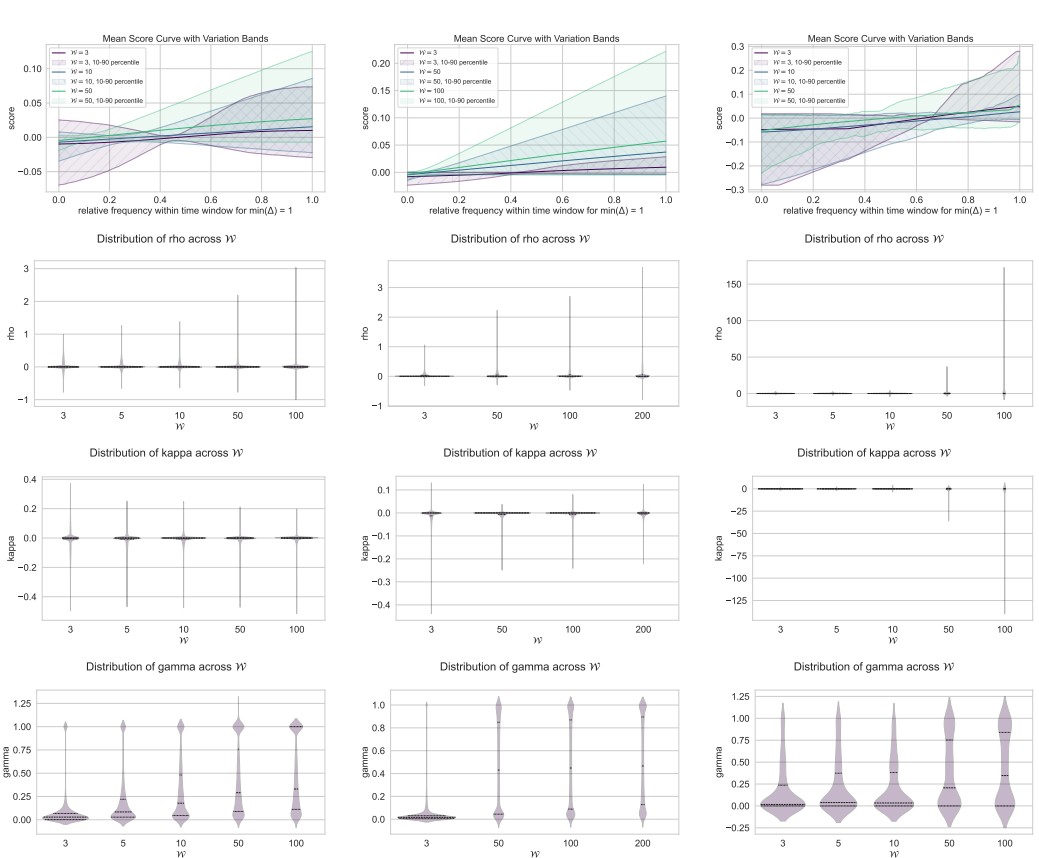

Figure 6: Frequency function $g$ and the effect of Learn Window Size $\mathcal{W}$ on learned parameters for datasets `ICEWS14`, `GDELT`, and `WIKI` (left to right). Note that the y-axis varies across datasets.

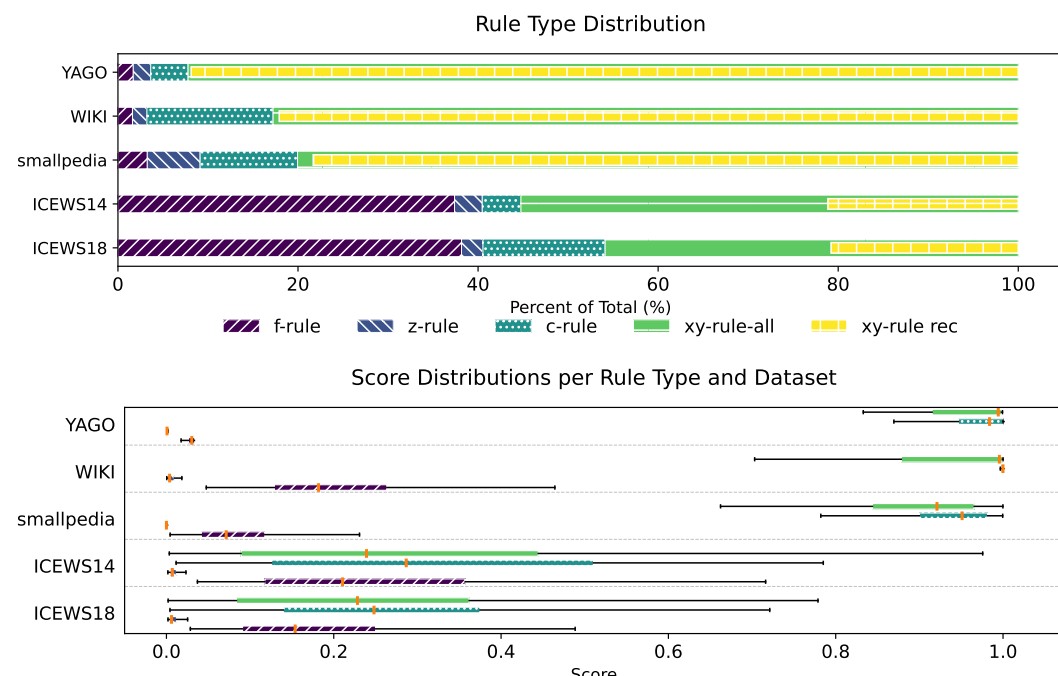

Figure 7: Distributions for the four rule types (top), and assigned scores for each rule type (bottom) for the highest predicted node and highest predicting rule for each test query.

### A.8.3 RULE TYPE DISTRIBUTIONS

Figure 7 shows the distributions of rule types (top) and the corresponding scores (bottom) for the highest-predicted node and its top-predicting rule across all test queries in the small and medium sized datasets.

The figure illustrates that the contributions of rule types vary significantly across datasets. For datasets based on Wikidata (`WIKI`, `smallpedia`) and YAGO (`YAGO`), which exhibit a high degree of recurrency (Gastinger et al., 2024b), xy-recurrency rules have the highest contributions. This is consistent with the ablation results in Table 2 (a), where CountTRuCoLa on `WIKI` achieves its best MRR using only recurrency rules. In contrast, for `ICEWS14` and `ICEWS18`, recurrency rules account for only around 20% of the contribution. Large contributions come also from non-recurrent xy-rules and f-rules, again reflecting the trends observed in Table 2 (a). Across all datasets, z-rules contribute minimally ($< 5\%$).

The score distributions in Figure 7 (bottom) show that xy- and c-rules achieve higher average scores for Wikidata- and YAGO-based datasets, whereas their average scores are substantially lower in `ICEWS14` and `ICEWS18`. The z-rules consistently have low median scores, as they primarily capture general distributions. The c-rules show the highest median scores for most datasets, which aligns with their specialized nature and ability to provide targeted, high-confidence predictions.

Overall, these distributions confirm our intuition: xy-rules contribute the most across datasets, but other rule types also play an important role in achieving high performance.

### A.8.4 RUNTIME

Table 8 reports runtimes in seconds for CountTRuCoLa on all datasets. We report total time (incl. loading the dataset, and necessary steps), and times for the individual steps reported in the main paper, i.e., creating the Examples $E$ for each rule, learning the parameters for each rule, rule application and aggregation, and evaluation. The runtimes were measured on a AMD EPYC 9474F 48-Core Processor. We did not use GPUs. Parallelization during rule application was handled via Ray, with a maximum of 20 concurrent threads.

For comparison, Table 9 shows the runtimes reported in Gastinger et al. (2024a) on the datasets presented in their study, and Table 10 shows runtimes for the method CognTKE on the datasets

Table 8: CountTRuCoLa runtimes for all datasets in seconds.

| | GDELT | YAGO | WIKI | ICEWS14 | ICEWS18 | smallp. | polecat | icews | wikidata |
|---|---|---|---|---|---|---|---|---|---|
| # rules | 134,574 | 8,140 | 350 | 53,553 | 1,010,307 | 1,221 | 1,024 | 167,670 | 12,050 |
| creation of examples [s] | 12,244 | 9 | 10 | 111 | 242 | 7 | 1,464 | 129,139 | 280 |
| parameter learning [s] | 5,670 | 11 | 0.4 | 366 | 7,743 | 1 | 594 | 2,221 | 18 |
| rule application [s] | 3,704 | 125 | 106 | 279 | 12,593 | 194 | 10,957 | 137,574 | 39,911 |
| evaluation [s] | 253 | 45 | 18 | 64 | 510 | 69 | 620 | 6335 | 2,637 |
| total time [s] | 21,926 | 191 | 143 | 825 | 21,149 | 280 | 13,681 | 276,269 | 43,029 |

Table 9: Inference time as well as total train and validation times as reported in Gastinger et al. (2024a) in seconds.

| Method | smallpedia | | polecat | | icews | | wikidata | |
|---|---|---|---|---|---|---|---|---|
| | Test | Total | Test | Total | Test | Total | Test | Total |
| EdgeBank$_{tw}$ | 2,935 | 5,810 | 46,629 | 94,475 | 311,278 | 600,929 | 5,445 | 8,875 |
| RecurrencyBaseline | 310 | 9,895 | 3,392 | 80,378 | 3,928 | 148,710 | - | - |
| RE-GCN | 165 | 3,895 | 1,766 | 45,877 | 6,848 | 114,370 | - | - |
| CEN | 331 | 14,493 | 2,726 | 77,953 | 8,999 | 202,477 | - | - |
| TLogic | 331 | 803 | 75,654 | 138,636 | 60,413 | 128,391 | - | - |

where we evaluated it. Since these results were obtained on different hardware[4], they are not directly comparable, but they provide a useful point of reference.

Table 9 shows that CountTRuCoLa's total runtimes for smallpedia and polecat are significantly lower than those of related work, despite not using a GPU. For icews, it achieves comparable runtimes to other methods, and for wikidata, it is the only method aside from the simple Edgebank heuristic that produces results. Compared to CognTKE, the strongest overall baseline in terms of MRR, Table 10 shows that CountTRuCoLa yields significantly lower total runtimes across all datasets. Although measured on different hardware, these results indicate that CountTRuCoLa is highly efficient in practice, requiring only CPU resources while achieving lower runtimes than GPU-accelerated baselines.

### A.8.5   HITS@1 AND HITS@3 RESULTS

Tables 11 and 12 report the results including Hits@1 and Hits@3 values.

The Hits@1 results do not provide a different impression from the MRR results presented in Table 1. Whenever CountTRuCoLa ranks first, second, or third in MRR among the compared models, it has the same position with respect to the Hits@1 score. The only exceptions are YAGO, where CountTRuCoLa moves from first to second place for Hits@1, and ICEWS18, where it moves from third to second place. The ranking of the Hits@3 values is also consistently within the top three positions among related work.

For the TGB2.0 datasets in Table 12, no Hits@1 and Hits@3 results were reported for related work. We thus compare our results only against CognTKE. On one, CognTKE achieves higher Hits@1

---

[4]Experiments in Table 9 were conducted on Nvidia A100, V100, V100SXM2, and RTX8000 GPUs with 4 CPU nodes (from AMD Rome, Milan, or Intel Skylake) per experiment, using up to 1056 GB of RAM, and Experiments in Table 10 were conducted on a NVIDIA RTX A6000 GPU with 48 GB VRAM with 4 CPU nodes, using up to 1024 GB of RAM.

Table 10: Test time as well as total train and validation times for CognTKE in seconds.

| | GDELT | | YAGO | | ICEWS14 | | smallpedia | | polecat | | icews | | wikidata | |
|---|---|---|---|---|---|---|---|---|---|---|---|---|---|---|
| | Test | Total | Test | Total | Test | Total | Test | Total | Test | Total | Test | Total | Test | Total |
| CognTKE | OOM | OOM | 78 | 8,287 | 36 | 6,080 | 711 | 63,300 | 7,548 | 417,600 | OOM | OOM | OOM | OOM |

Table 11: Model comparison including Hits@1 and Hits@3 on five datasets. OOM means Out Of Memory (40 GB GPU or 1056 GB RAM), OOT means Out Of Time (7 days), - means that no results have been reported. Best results are shown in bold and underlined font, second-best bold, third-best underlined.

| | GDELT | | | | YAGO | | | | WIKI | | | | ICEWS14 | | | | ICEWS18 | | | |
|---|---|---|---|---|---|---|---|---|---|---|---|---|---|---|---|---|---|---|---|---|
| | MRR | H1 | H3 | H10 | MRR | H1 | H3 | H10 | MRR | H1 | H3 | H10 | MRR | H1 | H3 | H10 | MRR | H1 | H3 | H10 |
| TRKG | 21.5 | 13.7 | 24.0 | 37.3 | 71.5 | 65.7 | 77.3 | 79.2 | 73.4 | 71.2 | 75.6 | 76.2 | 27.3 | 16.5 | 31.1 | 50.8 | 16.7 | 8.3 | 18.2 | 35.4 |
| xERTE | 18.9 | 12.7 | 21.1 | 32.0 | 87.3 | 84.2 | 90.3 | 91.2 | 74.5 | 70.3 | 78.6 | 80.1 | 40.9 | 33.0 | 45.5 | 57.1 | 29.2 | 20.9 | 33.5 | 46.3 |
| TANGO | 19.2 | 12.2 | 20.4 | 32.8 | 62.4 | 59.0 | 64.7 | 67.8 | 50.1 | 48.3 | 51.4 | 52.8 | 36.8 | 27.3 | 40.8 | 55.1 | 28.4 | 19.1 | 31.9 | 46.3 |
| Timetraveler | 20.2 | 14.1 | 22.2 | 31.2 | 87.7 | 84.6 | 90.9 | 91.2 | 78.7 | 75.2 | 82.0 | 83.1 | 40.8 | 31.9 | 45.4 | 57.6 | 29.1 | 21.3 | 32.5 | 43.9 |
| TiRGN | 21.7 | 13.6 | 23.3 | 37.6 | 88.0 | 84.3 | 91.4 | 92.9 | 81.7 | 77.8 | 85.1 | 87.1 | 44.0 | 33.8 | 49.0 | 63.8 | 33.7 | 23.2 | 38.0 | 54.2 |
| CognTKE | OOM | OOM | OOM | OOM | 90.6 | 88.1 | 92.9 | 93.2 | 83.2 | 80.0 | 86.0 | 87.3 | 46.1 | 36.5 | 51.1 | 64.5 | 35.2 | 25.2 | 39.9 | 54.7 |
| TLogic | 19.8 | 12.2 | 21.7 | 35.6 | 76.5 | 74.0 | 78.9 | 79.2 | 82.3 | 78.6 | 86.0 | 87.0 | 42.5 | 33.2 | 47.6 | 60.3 | 29.6 | 20.4 | 33.6 | 48.1 |
| RE-GCN | 19.8 | 12.5 | 21.0 | 33.9 | 82.2 | 78.7 | 84.2 | 88.5 | 78.7 | 74.8 | 81.7 | 84.7 | 42.1 | 31.4 | 47.3 | 62.7 | 32.6 | 22.4 | 36.8 | 52.6 |
| CEN | 20.4 | 13.0 | 21.8 | 35.0 | 82.7 | 78.8 | 85.2 | 89.4 | 79.3 | 75.5 | 82.4 | 84.9 | 41.8 | 31.9 | 46.6 | 60.9 | 31.5 | 21.7 | 35.4 | 50.7 |
| EdgeBank$_{tw}$ | 1.9 | 0.1 | 0.9 | 3.5 | 61.7 | 44.1 | 68.5 | 61.7 | 58.5 | 39.4 | 67.3 | 84.4 | 13.5 | 3.2 | 14.1 | 34.2 | 7.2 | 1.5 | 6.3 | 17.9 |
| Rec.B ($\psi_\triangle \xi$) | 24.5 | 16.4 | 26.8 | 39.8 | 90.9 | 89.0 | 92.8 | 93.0 | 81.4 | 76.9 | 85.7 | 87.1 | 37.4 | 29.9 | 41.2 | 51.5 | 28.7 | 20.8 | 32.3 | 43.6 |
| CountTRuCoLa | 23.8 | 15.4 | 26.3 | 40.3 | 90.9 | 88.8 | 92.9 | 93.2 | 82.7 | 79.4 | 85.7 | 86.6 | 45.0 | 36.0 | 49.8 | 62.0 | 32.8 | 23.5 | 36.9 | 51.0 |

Table 12: Model comparison including Hits@1 and Hits@3 on the TGB2.0 benchmark datasets. OOM means Out Of Memory (40 GB GPU or 1056 GB RAM), OOT means Out Of Time (7 days), - means that no results have been reported. Best results are shown in bold and underlined font, second-best bold, third-best underlined.

| | smallp | | | | polecat | | | | icews | | | | wikidata | | | |
|---|---|---|---|---|---|---|---|---|---|---|---|---|---|---|---|---|
| | MRR | H1 | H3 | H10 | MRR | H1 | H3 | H10 | MRR | H1 | H3 | H10 | MRR | H1 | H3 | H10 |
| CognTKE | 53.4 | 44.0 | 60.2 | 70.1 | 28.7 | 20.1 | 32.7 | 45.5 | OOM | OOM | OOM | OOM | OOM | OOM | OOM | OOM |
| TLogic | 59.5 | - | - | 70.7 | 22.8 | - | - | 37.8 | 18.6 | - | - | 30.1 | OOT | OOT | OOT | OOT |
| RE-GCN | 59.4 | - | - | 68.7 | 17.5 | - | - | 29.2 | 18.2 | - | - | 33.1 | OOM | OOM | OOM | OOM |
| CEN | 61.2 | - | - | 70.5 | 18.4 | - | - | 32.3 | 18.7 | - | - | 33.4 | OOM | OOM | OOM | OOM |
| EdgeBank$_{tw}$ | 35.3 | - | - | 56.6 | 5.6 | - | - | 11.9 | 2.0 | - | - | 5.8 | 53.5 | - | - | 59.6 |
| Rec.B ($\psi_\triangle \xi$) | 60.5 | - | - | 71.6 | 19.8 | - | - | 31.7 | 21.1 | - | - | 32.4 | OOT | OOT | OOT | OOT |
| CountTRuCoLa | 64.4 | 59.5 | 68.7 | 71.7 | 25.6 | 17.8 | 29.1 | 40.8 | 21.4 | 15.7 | 24.1 | 32.1 | 60.9 | 59.8 | 61.6 | 62.8 |

and Hits@3 scores, on another CountTRuCoLa achieves the highest results, and on two datasets, CognTKE did not produce results due to OOM errors.

### A.8.6 VARIANCE ACROSS REPETITIONS

Table 13 reports test results across five repetitions for two selected datasets. We observe that the variance is less than $0.002$ for all test metrics and datasets, which is small considering that the MRR and Hits@10 range from $0\%$ to $100\%$. Compared to embedding-based methods, which may introduce randomness through factors such as random initialization or noise injection, our approach has fewer potential sources of variability: (i) sampling candidates for the examples of $c$-rules, (ii) computing the parameters that optimize the temporal confidence functions, and (iii) assigning random order to tied ranks during evaluation.

### A.8.7 SIGNIFICANCE TESTS

We conduct a Chi-squared test to assess whether the observed differences in Hits@10 scores between methods are statistically significant. Since Hits@10 is a binary metric (a prediction is either in the

Table 13: Test results on WIKI and ICEWS14 (5 runs, mean, variance).

| | WIKI | | ICEWS14 | |
|---|---|---|---|---|
| | MRR | H10 | MRR | H10 |
| | 82.6809 | 86.5592 | 44.9916 | 62.0201 |
| | 82.6615 | 86.5568 | 44.9828 | 62.0404 |
| | 82.5704 | 86.5647 | 45.0014 | 62.1083 |
| | 82.6181 | 86.5624 | 44.9864 | 62.0336 |
| | 82.5992 | 86.5584 | 44.9685 | 62.0201 |
| Mean | 82.6260 | 86.5603 | 44.9862 | 62.0445 |
| Variance | 0.002038 | 0.000010 | 0.000146 | 0.001348 |

Table 14: Chi-squared test for all datasets. We test for ($df = 1$, $p < 0.001$). and compare Count-TRuCoLa with the best performing state of the art method regarding Hits@10.

| Dataset | CountTRuCoLa H10 | Best Baseline H10 | Best Baseline | $\chi^2$ | Significant? |
|---|---|---|---|---|---|
| **Group 1: CountTRuCoLa best method** | | | | | |
| GDELT | 40.3 | 39.8 | Rec. B. | 31.78 | yes |
| YAGO | 93.2 | 93.2 | CognTKE | 0 | no |
| smallpedia | 71.7 | 71.6 | Rec. B. | 0.40 | no |
| wikidata | 62.8 | 59.6 | Edgebank | 6204.43 | yes |
| **Group 2: CountTRuCoLa not best method** | | | | | |
| WIKI | 86.6 | 87.3 | CognTKE | 27.25 | yes |
| ICEWS14 | 62.0 | 64.5 | CognTKE | 19.81 | no |
| ICEWS18 | 51.0 | 54.7 | CognTKE | 272.19 | yes |
| icews | 32.1 | 33.4 | CEN | 1784.58 | yes |
| polecat | 40.8 | 45.4 | CognTKE | 2297.88 | yes |

top 10 or not), we treat the predictions as categorical outcomes: *hit* and *miss*. For a dataset, we have $2 \times N$ samples (where $N$ is the number of test quadruples), accounting for inverse relations.

For example, if a model achieves a Hits@10 score of 0.6 on a test set with 100 quadruples, this corresponds to $0.6 \times 200 = 120$ *hit* predictions and 80 *miss* ones.

We compare CountTRuCoLa with the best-performing state of the art method regarding Hits@10 under the following null hypothesis:

**H$_0$**: Prediction correctness (i.e., *hit* vs. *miss*) is independent of the method used.

For examples, for the GDELT dataset, the Chi-squared test indicates a significant association between prediction correctness and method ($\chi^2 = 31.8$, $df = 1$, $p < 0.001$), suggesting that the improvement in Hits@10 by CountTRuCoLa over the baseline is statistically significant. Based on $p < 0.001$, in total, for the 4 datasets, where CountTRuCoLa has higher Hits@10 scores than the others, for 2 of them the Chi-squared tests suggests a statistically significant improvement. For the 5 datasets, where CountTRuCoLa has lower Hits@10 scores than the other methods, for 4 of them the Chi-squared test suggests a statistically significant difference.

### A.8.8 MEMORY LIMITS

The upper memory limits set in SLURM for each dataset (using 20 parallel processes) are as follows:

- Small datasets (WIKI, ICEWS14, YAGO, tkgl-smallpedia): 20 GB
- polecat: 160 GB
- ICEWS18: 300 GB
- gdelt: 350 GB
- wikidata: 200 GB
- tkgl-icews: 500 GB

Please note that these are upper limits, not actual memory usage. Additionally, reducing the number of parallel processes during the application phase will decrease memory consumption at the cost of increased runtime.

### A.9 DETAILS ON THE EXPLANATION TOOL

In the following, we provide additional details on the explanation tool introduced in Section 6. The code for the explanation tool is also published in the provided repository, along with notebooks demonstrating its use.

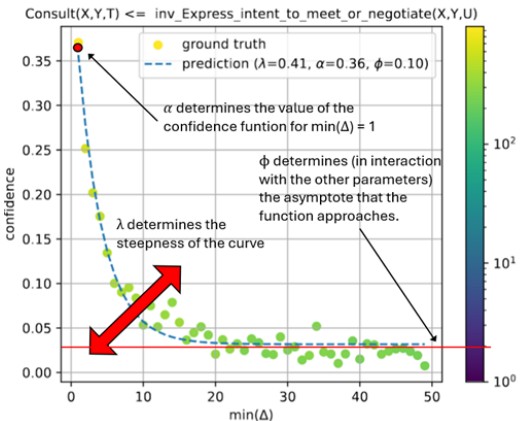

Figure 8: Examples $E$ (points) and predicted confidence curves (blue lines) $f$ for a rule $r$. Colors indicate the number of samples in $E$. The text explains the role of each parameter $\lambda, \alpha, \phi$.

**Pre-Analysis Features**   Before generating explanations, the tool helps gain an overview of datasets and models. It provides (i) dataset statistics like relation distributions, and number of triples over time, computes (ii) a fine-grained evaluation with metrics like MRR per relation and timestep, and conducts (iii) a ranking comparison that identifies quadruples where two methods differ significantly.

For example, the ranking comparison showed that on the dataset ICEWS14, CountTRuCoLa consistently outperforms RE-GCN on the relation *Consult*, suggesting that simple temporal rules capture this relation comparatively well.

**Input and Workflow**   The tool takes as input (i) a rule set (learned or user-defined), (ii) an optional configuration (e.g., aggregation function, window size), and (iii) queries to explain. It then applies rules with CountTRuCoLa, records applied rules and scores, conducts an evaluation and generates visual explanations.

For instance, selecting all *Consult* queries where CountTRuCoLa outperforms RE-GCN enables targeted investigation. This allows researchers to inspect which rule patterns explain correct forecasts and where neural models may miss such dependencies.

**Output**   For each query, the tool produces a structured explanation that includes the predicted node and score, the rules that contributed (with their parameters), visualizations of temporal confidence functions, and the frequency/recency of supporting quadruples. In addition, the tool outputs evaluation scores (MRR, Hits@k) for the quadruples of interest, allowing fine-grained comparison of settings.

## A.10   EXAMPLES FOR LEARNED RECENCY FUNCTIONS

As explained in Section 4.2, every parameter in CountTRuCoLa has a fixed and interpretable role. Figure 8 illustrates the role of each parameter $\lambda, \alpha, \phi$ in the recency function $f$ for an individual rule. Figure 9 provides three concrete examples of rules that have learned different parameters, along with the corresponding sets of examples $E$ that were used to learn these curves. Together, the figures highlight that different rules indeed need different parameters to describe their behaviour.

For example, the first rule exhibits a steep decay, with its confidence approaching zero as $\min(\Delta)$ increases. In contrast, the second rule requires a relatively shallow decay. The third rule does not converge to zero and is supported by positive examples at higher $\min(\Delta)$.

These observations align with our Ablation Study in Section 5.2, Table 2(c), which demonstrates that learning rule-specific parameters improves performance. In summary, the study confirms that different rules indeed require different confidence-function shapes, and modeling them individually is beneficial.

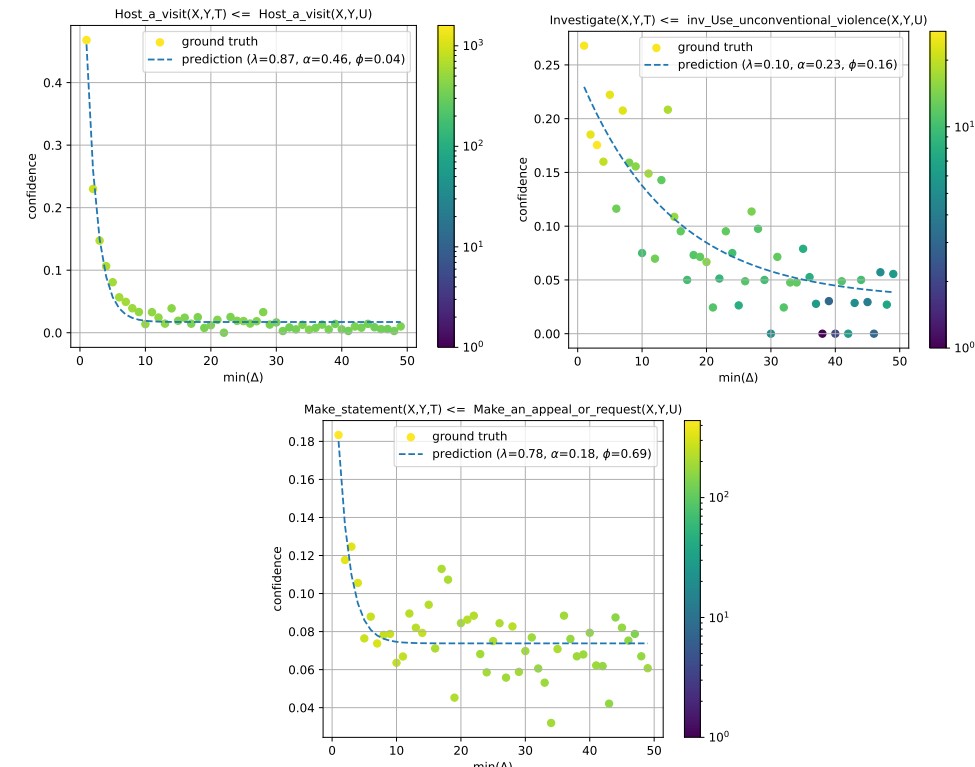

Figure 9: Examples $E$ (points) and predicted confidence curves (blue lines) $f$ for three rules. Colors indicate the number of samples in $E$.

### A.11 LLM USAGE

We used a large language model (ChatGPT, GPT-4 and GPT-5 by OpenAI) during the preparation of this manuscript. Its role was limited to language-level assistance, including polishing sentence structure and grammar, reformulating paragraphs for clarity, suggesting synonyms to reduce repetition, and smoothing transitions between sentences.

The model was not used to generate research ideas, experimental designs, data analyses, results, or scientific claims. All substantive content, technical contributions, and conclusions are the work of the authors.

