# OpenReview forum: "CountTRuCoLa: Rule Learning for Explainable Temporal Knowledge Graph Forecasting"
_ICLR.cc/2026/Conference — ICLR 2026 Conference Withdrawn Submission_

### Official Review · Reviewer_jc2c · 2025-10-30

**Soundness:** 3
**Presentation:** 3
**Contribution:** 2
**Rating:** 4
**Confidence:** 3

**Summary:**

This paper presents CountTRuCoLa, an explainable rule-based method for forecasting future relationships in temporal knowledge graphs. Inspired by a repetition-based baseline, it uses four simple rule types (e.g., repeating entity-relation patterns, cross-relation links with common entities) and a confidence score that considers how recent and frequent past events were. It combines top rules’ predictions to rank results, and tests on nine datasets show it performs well, avoids memory/time issues with large graphs, runs efficiently on CPUs, and lets users trace predictions to specific rules and past events—proving simple, explainable methods can match complex neural models.

**Strengths:**

1. The proposed method is fully interpretable, allowing every prediction to be traced back to specific temporal rules and observed facts—an important contribution to transparency in TKG forecasting.

2. Despite its rule-based simplicity and CPU-only implementation, the model achieves competitive accuracy and superior scalability compared to complex neural or reinforcement learning models.

3. The authors conduct extensive experiments on nine datasets, including ablation studies that clearly demonstrate the contribution of different rule types and confidence components.

**Weaknesses:**

1. The paper mainly extends existing rule-based and recurrence-driven approaches without introducing new ideas in explainability. It focuses on weighting known temporal patterns via confidence functions rather than advancing interpretable modeling, such as complex or compositional rule reasoning.

2. The work lacks thorough ablation and sensitivity analyses of the confidence function. The model relies on parameter tuning across numerous rule instances rather than uncovering consistent or generalizable temporal regularities, weakening its explainability claim.

3. The experimental evaluation is not sufficiently up-to-date. Only one 2024 baseline method, Rec.B, is included, while recent baselines are missing and should be compared.

4. The paper does not report the Hits@1 and Hits@3 results, which are widely used evaluation metrics in TKGR research.

**Questions:**

See the weakness.

---

> ### Author Response · Authors · 2025-11-21
> **Response to Reviewer 3 (jc2c)**
>
> We thank the reviewer for taking time to assess our paper and offering valuable suggestions. We are happy that the reviewer considers full interpretability an important contribution to transparency in TKG forecasting.
> Following the feedback, we updated our paper, and provide detailed answers below.
>
> ### W1 (missing novelty):
> We acknowledge the concern on limited novelty in terms of explainability mechanisms.
> We are aware that our paper contains less novel and sophisticated ideas compared to many other submissions. We agree that incorporating, as proposed, compositional rule reasoning, could potentially improve results further.
> However, doing so would shift the message of this work.
> The aim of our paper is to demonstrate that **a simple model that learns and combines confidence functions over basic temporal patterns can achieve performance competitive with, and in some cases superior to, state-of-the-art approaches.**
>
> While the Rec.B. and TLogic share ideas with our method, their results are substantially lower. From these papers you cannot draw the conclusion that models based on such simple patterns can match the performance of more complex architectures.
>
> We believe that our findings are important for the TKG community.
> They highlight that strong predictive performance does not necessarily require complex models.  Our paper clarifies how far one can get with simple approaches before adding more complex mechanisms.
> Our results are important since they are not in line with what most researchers would expect as performance of such a restricted and interpretable model.
> ### W2 (full explainability claim):
> Reviewer jc2c mentions as a problem "(I) parameter tuning on (II) numerous rule instances rather than uncovering consistent or generalizable temporal regularities, weakening its explainability claim".
> This can be decomposed in (II) the fact that we learn thousands of rules and (I) the fact that each rule has 6 parameters.
>
> **Aspect (I):**
>
> As each (xy-and c-)rule deals with temporal dependencies, we must learn a function instead of a fixed confidence value. A fixed value would be one parameter per rule. Instead we have 6 params that describe two functions. The first, $f$, is an exponential function. We depict the role of each of the 3 parameters in Figure 8 in the Appendix. The second, $g$, is a straight line (2 parameters) with an additional parameter to dermine where to cut this line. Each of the 6 parameters has a clear meaning.
>
> **Aspect (II):**
> Indeed, CountTRuCoLa learns thousands of rules. However, each prediction can be derived from few rules only. Even if many rules result in the same candidate prediction, only the top rules contribute to the final score, according to the noisy-or with decay factor. Thus, the candidate score and its ranking position is usually determined by few (often $<3$) rules.
> In the example in Figure 2 we know exactly that Evangelis V. is the top ranked candidate due to the rule with confidence score 0.169, how the confidence function looks, and which quadruple caused the prediction.
>
> **Ablation and Sensitivity analyses**: To evaluate the need for learning parameters, we **added a new Ablation study** in Table 2$($c$)$ (Section 5.2). It shows that learning rule-specific parameters is beneficial. Different rules require differently parametrized confidence functions. Examples can be seen in the Appendix in Figure 9.
>
> In summary: We indeed **need to learn the parameters**, but **this does not restrict the quality of CountTRuCoLa's explanations**.
>
> ### W3 (outdated evaluation):
> We appreciate the reviewer’s concern regarding the recency of the baselines in our evaluation. However, we would like to clarify that **our comparison does include recently proposed methods**. In particular, **CognTKE**, which appears in our results table, was published in 2025, and therefore provides an up-to-date baseline beyond Rec.B (2024).
>
> Furthermore, in **Appendix 5.1** we explain in detail why several other recent models were not included in our comparison. E.g., Infer (2025) reports results only under the “best” ranking protocol, which is recognized to inflate scores and therefore prevents a fair comparison. We were unable to re-evaluate Infer, as necessary specifications for pretraining the required Complex model are missing.
>
> If there are additional recent methods that we may have overlooked and that are not listed in Appendix 5.1, we would be grateful if the reviewer could point them out.
>
> ### W4 (Hits@1 and Hits@3)
> Thank you for pointing this out. **We now added results with Hits@1 and 3 in Table 11 and 12.**
>
> In Summary, the Hits@1 results do not provide a different impression from the MRR results. Whenever CountTRuCoLa ranks first, second, or third in MRR among the compared models, it has the same position with respect to the Hits@1 score (for two datsets, positions have flipped). The ranking of the Hits@3 values is also consistently within the top three positions among related work.

---

> > ### Author Response · Authors · 2025-11-28
> > **Official Comment by Authors**
> >
> > Thank you once again for taking the time to review our paper. As the discussion period is approaching its end, we would like to confirm whether our responses have sufficiently addressed your concerns. Moreover, we are more than willing to provide additional explanations if you have any further questions.

---

### Official Review · Reviewer_2EQa · 2025-10-30

**Soundness:** 3
**Presentation:** 3
**Contribution:** 2
**Rating:** 6
**Confidence:** 4

**Summary:**

This paper presents a fully explainable temporal knowledge graph forecasting method that learns four simple types of temporal rules, leveraging a confidence function based on recency and frequency; extensive evaluation on nine datasets shows that the approach delivers competitive and often superior performance compared to state-of-the-art models, while maintaining interpretability in its predictions.

**Strengths:**

1. The rule-based approach enables explicit explanations for each prediction.

2. The paper's motivation is clear and the writing is sufficiently detailed. Each parameter in the confidence function has a clear, intuitive interpretation.

3. The experiments are comprehensive, validating the method’s effectiveness on nine datasets.

**Weaknesses:**

1. Although the paper defines four rule types, the definitions and examples appear to lack consideration of multi-hop rules.

2. The z-rules and f-rules are static and do not incorporate any temporal dynamics, which may be a poor fit for datasets where the frequency distributions of entities or relations change significantly over time

3. In Equation (12), the functional form of $\(f_r\)$ appears ad hoc, lacking derivation from first principles or probabilistic models. Its selection is not justified, and it is unclear why this form is preferable to simpler exponential decay.

4. The paper introduces numerous hyperparameters, both trainable and non-trainable. However, the necessity of so many parameters remains questionable due to a lack of supporting experiments.

5. Although experiments are conducted on multiple datasets, the authors do not report Hits@1 and Hits@3 results. These metrics, particularly Hits@1, can be critically important in some cases.

**Questions:**

See Weakness

---

> ### Author Response · Authors · 2025-11-21
> **Response to Reviewer 2 (2EQa)**
>
> We thank the reviewer for taking time to assess our paper and offering valuable suggestions. Following your feedback, we updated our paper accordingly, and provide detailed answers on each point below.
>
> ### W1 (no muli-hop rules):
> It is correct that CountTRuCoLa does not support multi-hop rules. Our objective was to design a deliberately simple and fully interpretable model based solely on single-atom dependencies. This allows us to isolate and better understand which temporal regularities are already sufficient for strong predictive performance.
>
> Despite this simplicity, CountTRuCoLa achieves competitive performance across diverse datasets, outperforming many existing approaches and even reaching state-of-the-art results on some datasets. This shows that, for current datasets, much of the predictive gain attributed to multi-hop capabilities can be captured with simple, single-step dependencies. Our results are in contrast with assumptions made in several prior works which motivate multi-hop rules as important for accurate predictions (e.g., TLogic considers rules up to length 3).
>
> We believe it is valuable to see that a deliberately limited, interpretable model can perform that strongly, which would probably not be expected by most researchers working in that field.
>
>
> ### W2 (static z-rules and f-rules do not incorporate temporal dynamics):
> It is true that z-rules and f-rules do not support temporal dynamics. This means that, from the perspectice of a z or f-rule, an entity that was involved in many quadruples at early time points is treated the same as an entity that was involved in many quadruples at later, more recent time points.
> In a first version of CountTRuCoLa we tried to take this difference into account by assigning heigher weights to more recent observations. We conducted preliminary experiments using different weighting schemes. However, we could not observe a significant improvement and thus decided at an early stage not to include this weighting mechansim to favor simplicity over complexity.
>
> ### W3 (functional form of $f$ ad hoc):
> **We have added a motivation for our choice of the recency function $f$ to the paper in Section 4.2**:
> > Using an exponential decay function to model temporal decay was inspired by TLogic, which uses such a function to express temporal confidence. However, several modifications were necessary to capture phenomena that cannot be represented in TLogic’s formulation.
> > First, whereas TLogic employs a single fixed decay factor for all rules, we learn a separate set of parameters for each rule. This allows the model to represent diverse temporal behaviours, such as rules that lose relevance quickly and others that remain informative over longer time spans.
> > Second, we introduce $\phi_r$ to control the asymptotic lower bound of $f_r$. Empirically, we observed that some rules exhibit an initial drop in relevance but keep a stable level of influence afterwards. This behaviour cannot be captured by a fixed exponential decay without a learnable lower bound.
>
> An example, where $\phi_r$ is beneficial is the rule
> `Make_statement(X,Y,T) <= Make_an_appeal_or_request(X,Y,U)`
> shown at the bottom of the new Figure 9 in the Appendix. We have also visualized the purpose of all parameters in the new Figure 8.
>
> ### W4 (hyperparameters really required?):
> Thank you for raising this.
> **We now added studies to justify the parameters and hyperparameters**.
>
> * The new ablation study in Table 2$($c$)$ (Section 5.2) evaluates whether learning the confidence-function parameters is necessary by fixing different subsets of the parameters. The study shows that learning rule-specific parameters is important, as different rules require different confidence-function shapes.
> * The new Figure 9 shows examples for three rules that learn different $f$ shapes.
>
> * The new Appendix A.7 analyses Hyperparameter Sensitivity
>     * Figure 4 shows the effect of selected hyperparameters on the test MRR, showing that indeed MRRs do vary significantly when changing the hyperparameters.
>     * Table 6 compares the MRR when hyperparameters are selected based on validation performance (as reported in Appendix A.6 and Table 5) versus using default hyperparameter values. Results show that hyperparameter selection generally improves performance. For example, the difference for GDELT ranges from 20.3 to 23.8, whereas other datasets, such as YAGO, are relatively stable, with difference of 0.1.
>
>
> ### W5 (hits@1 and hits@3)
> Thank you for pointing this out. **We now added results with Hits@1 and 3 in Table 11 and 12.**
>
> In Summary. the Hits@1 results do not provide a different impression from the MRR results presented so far. Whenever CountTRuCoLa ranks first, second, or third in MRR among the compared models, it
> has the same position with respect to the Hits@1 score (for two datsets, positions have flipped). The ranking of the Hits@3 values is also consistently within the top three positions among related work

---

> > ### Author Response · Authors · 2025-11-28
> > **Official Comment by Authors**
> >
> > We truly appreciate your time in reviewing our paper, thank you once again. As the discussion period comes to an end, we would like to confirm whether our responses  have effectively addressed your concerns. If you have any remaining questions, we are more than willing to provide further explanations.

---

### Official Review · Reviewer_Qwtd · 2025-10-30

**Soundness:** 3
**Presentation:** 3
**Contribution:** 3
**Rating:** 4
**Confidence:** 3

**Summary:**

The paper presents CountTRuCoLa, an interpretable rule-based framework for temporal knowledge graph forecasting. The method learns four families of temporal rules and equips each rule with a parametric temporal confidence that jointly captures recency and frequency of supporting events. At inference, predictions from multiple rules are fused via a truncated, decayed noisy-OR aggregator over the top-H rule confidences. The approach targets single-step extrapolation, runs efficiently on CPU, and ships an explainer that attributes each prediction to concrete rules and their fitted confidence curves. Experiments across a broad suite of benchmarks report competitive or state-of-the-art MRR together with favorable runtime and memory behavior.

**Strengths:**

1.	Interpretability and transparency – Every prediction can be traced back to concrete symbolic rules and corresponding confidence curves. The use of four rule types enables fine-grained temporal reasoning while retaining human interpretability.
2.	Principled temporal modeling – The confidence function decomposes temporal influence into a recency component and a frequency component, each with distinct, interpretable parameters (α, λ, ϕ, ρ, κ, γ). This formulation clearly captures short- and long-term dynamics.
3.	Sound aggregation mechanism – The top-H decayed noisy-OR aggregator is a reasonable balance between simplicity and partial dependence modeling. Its hyperparameters (H, D) are explicitly defined and empirically validated.
4.	Comprehensive empirical evaluation – Nine datasets, extensive ablation studies, and runtime analyses are provided. CountTRuCoLa achieves strong predictive accuracy on both small and large graphs without GPU reliance.
5.	Explainability tool – The accompanying explanation interface visualizes rule activations, enabling comparison with neural baselines and facilitating post-hoc analysis of temporal dependencies.
6.	Reproducibility and documentation – Code, datasets, and appendices (A.1–A.9) offer complete transparency regarding data collection, hyperparameter tuning, and runtime environments.

**Weaknesses:**

1.	Lack of formal definition for “explainability” and rule faithfulness.
The paper claims interpretability through rule-level explanations and visualized confidence curves (Sec. 6), yet does not quantitatively measure whether these explanations are faithful to the prediction process. There is no evaluation of how rule confidence aligns with contribution scores or ground-truth relevance. A simple correlation or fidelity metric would help validate that interpretability is not merely descriptive.
2.	Counterfactual and causal reasoning terminology ambiguity.
Although the title and abstract emphasize “reasoning,” the framework is purely observational—confidence estimation depends only on empirical Δ-lag statistics (Sec. 4.2). There is no notion of intervention, counterfactual simulation, or temporal causation. Clarifying the term “reasoning” to mean “pattern-based temporal inference” would avoid conceptual confusion.
3.	Two-stage confidence fitting without sensitivity analysis.
In Eq. (14), the confidence function is decomposed into recency and frequency components fitted sequentially. This procedure lacks quantitative analysis of whether the order of fitting (recency-first) affects the balance between short- and long-term contributions. The absence of a sensitivity or ablation study limits understanding of parameter robustness.
4.	Simplifying assumptions in the noisy-OR aggregation.
The truncated decayed noisy-OR (Sec. 4.3) aggregates top-H rule confidences under an implicit conditional independence assumption. The paper acknowledges this simplification (last paragraph of Sec. 4.3) but does not empirically test its effect—e.g., whether correlated rules inflate probability estimates. A diagnostic example or comparison with a logistic combiner could strengthen this section.
5.	Limited evaluation horizon.
Experiments (Sec. 5) only cover single-step extrapolation. While this setting is common in temporal KGs, the paper does not analyze how rule confidence behaves under longer horizons or sparser timestamps, leaving uncertainty about the model’s stability for multi-step forecasting.

**Questions:**

1.	Confidence function shape: In Fig. 4, learned confidence curves exhibit various non-monotonic shapes. Could you characterize the conditions under which these curves are monotonic or peaked, and how such shapes affect prediction reliability?
2.	Sensitivity to H and decay parameters: The noisy-OR aggregator uses truncation parameter H and decay D (Sec. 4.3). Have you explored how varying these parameters impacts both MRR and rule interpretability? For instance, does larger H lead to overcounting correlated rules?
3.	Rule family contributions: The paper defines four rule types (xy, c, z, f). Could you report which families contribute most to top-ranked predictions across datasets? An aggregated table of rule coverage or confidence variance would clarify their relative importance.
4.	Temporal granularity: Since datasets such as ICEWS and GDELT have different time resolutions, how does the choice of window W affect the fitted recency decay parameters? Is there evidence that these parameters are transferable across datasets?
5.	Scalability breakdown: Can you provide more detailed timing results—such as per-dataset CPU time for rule extraction and fitting—to better substantiate the “CPU-efficient” claim?

---

> ### Author Response · Authors · 2025-11-21
> **Response to Reviewer Qwtd (1/2)**
>
> We thank Reviewer Qwtd for taking time to assess our paper and offering valuable suggestions. We are especially happy about the positive remarks related to the interpretability, even though there were also several critical comments. This is indeed one of the most important points that distinguishes our work from other approaches.
> We address each point below.
>
> ### W1 (formal definition for “explainability” & rule faithfulness):
> We **now clarified the term interpretability in Section 4.4**. Moreover, we realized that the use of the word explainability is misleading and thus replaced “explainable” with “interpretable” in the title and text. Thank you for pointing out that our paper was unclear w.r.t. that issue!
>
> We would like to emphazise that the explanatory structure is identical to the computation performed by the model, making CountTRuCoLa's prediction process (from input triples, to rule activations, confidence values, to the final prediction) transparent and inherently interpretable. This requires no post-hoc auxiliary explanation method and makes explanations inherently faithful.
>
> As pointed out by the reviewer, we had not analysed whether the score from a confidence function correlates with the probability that the prediction made by that rule is true.
> Thus, we now **added a detailed analysis in Appendix A.8.1, including Figure 4**
> We collected for each test query all predicted candidates and their aggregated confidence scores and ground-
> truth labels (true or false). We grouped the candidates in 10 confidence bins (i, i+0.1] with i in {0, 0.1, 0.2, …, 0.9}. For each bin, we computed the fraction of correct predictions.
>
>
> As an example, the upper plot in Figure 4 is based on H=1, i.e. if a candidate is generated by several rules, it only uses the highest confidence. The plot shows a strong correlation between accuracy and score, even though the predicted score overestimates the ground truth probability. **We added a detailed analysis in Appendix A.8.1.**
>
> ### W2 (emphasis on “reasoning” too strong):
> We could not detect any occurrence of the word “reasoning” to describe our approach in our paper. The impression that we emphasize reasoning, seems to be caused by the use of “explainability“. For that reason, and as argued above, we have changed the title and replaced “explainable” by “interpretable”. Moreover, we added Section 4.4 to explain why we refer to our method as interpretable. The notion of explainability was indeed misleading as it suggests that we need to use reasoning to generate explanations.
>
> ### W3 (two-stage confidence fitting)
> We agree that we did not provide a formal analysis of whether the order of fitting affects the balance between recency and frequency contributions.
>
> The recency-first order was chosen based on ablation studies (Table 2b), where the recency component $f$ alone outperforms the frequency component $g$ alone. Accordingly, we use $g$ as a complementary component to improve predictions of $f$.
>
> We also explored optimizing all 6 parameters jointly. This did not improve accuracy but increased traintime. The two-stage procedure provides competitive performance while remaining computationally efficient.
>
> ### W4 (simplifying assumptions in noisy-OR aggregation):
> Based on your suggestion, **we now tested the effect of the conditional independence assumption in the truncated decayed noisy-OR aggregation**. Specifically, we analysed how the number of top rules H and the decay parameter D influence both ranking performance (MRR, Hits@1) and the relationship between predicted confidence values and empirical correctness.
>
> 1. **Effect on MRR performance (Appendix A.7)**
> For most datasets:
> * performance improves with larger H, with no substantial improvements between H=10 and H=50. This suggests, that assuming independence up to a certain degree is valuable.
> * intermediate D values (0.5≤D≤0.8) yield the best performance, reflecting that some rules are partially correlated.
>
> 2. **Effect of aggregation on probability estimates (A.8.1)**
> * First, we study over- and underestimation across confidence bins for different aggregation strategies. E.g. on ICEWS14 the predicted scores by our decayed noisy-OR strategy most closely match empirical accuracies, and a monotonic relationship between confidence bins and accuracy is preserved
> * Second, we examine Top-1 predictions. Max-aggregation underestimates confidence, while noisy-or (H=100) clearly overestimates it, Using the decay factor (H=10, D=0.8) produces a closer match between predicted scores and Hits@1 and yields the highest Hits@1 among the tested configurations.
>
> These results show that the decay factor mitigates overcounting from correlated rules
>
> ### W5 (no multi-step forecasting):
> As the reviewer notes, it would indeed be interesting to study how rule confidence behaves over longer horizons or with sparser timestamps. However, following recent works (e.g., TGB2.0), in this paper, we focused on the single-step setting.

---

> > ### Author Response · Authors · 2025-11-21
> > **Response to Reviewer Qwtd (2/2)**
> >
> > ### Q1 (confidence function shape)
> >
> > We are not sure, if we correctly understand this question. If we are wrong, we would be grateful to hear a claryfing follow-up question.
> >
> > In Figure 1 (we think, this the figure that the reviewer refers to), the learned functions $f$ and $g$ (blue) are both monotonic: $f$ is exponential $g$ is linear. The question might be related to the datapoints (dots from yellow to blue) from which the functions have been learned.
> > E.g., in the left plot, a green dot at $x=30$ summarizes ~100 samples whose closest time distance to a firing quadruple was 30. About 3% of those samples were correct.
> > The observed datapoints form a zig zag patetrn, likely due to noise, which increases when the number of samples in a group is small.
> >
> > For $g$, some frequency groups contain only 1–2 samples (e.g., frequency 0.32), while others contain hundreds (e.g., frequency 0.02). This imbalance can make the observations appear non-monotonic, even though the effect is likely caused by the small-sample noise
> > Because groups with very few samples do not meaningfully affect prediction reliability, our aim is simply to fit a line that captures the overall monotonic trend.
> >
> > ### Q2 (sensitivity to H and D)
> > We **added several experiments** related to these questions **in Appendix A.7 and A.8**. A summary of the most important insights is in our reply to W4.
> >
> > ### Q3 (rule family contributions):
> > In Ablation study (a) (Table 2, Section 5.2) we already presented partial answers: xy-recurrency rules alone achieve the highest MRR, outperforming other rule types. For WIKI, they are sufficient, while for ICEWS14, top performance requires combining all rule types.
> >
> > We now additionally **report further results on rule type and confidence distributions in Figure 7, Appendix A 8.3**.
> >
> > Overall, rule-type contributions vary strongly by dataset.
> > Wikidata-datasets are dominated by xy-recurrency rules, whereas in ICEWS datasets, large contributions come from non-recurrent-xy- and f-rules.
> > Confidence distributions show that xy- and c-rules have much higher average scores for Wikidata-datasets than ICEWS-datasets.
> > c-rules exhibit the highest median confidence across most datasets, consistent with their specialized nature.
> >
> > We refer to the paper form more details.
> >
> > ### Q4 (temporal granularity, window size):
> > We **added a new section to the Appendix (A.8.2)** that analyses this question.
> >
> > In summary, for the recency parameters we find that:
> >
> > * The parameter values of GDELT (15 mins) & ICEWS14 (daily) exhibit similar trends as window size W increases: Larger windows tend to reduce both α and ϕ, while the smallest λ values appear at intermediate window sizes. This results curves that start lower but decay less steeply. Some rules learn sharply decaying recency curves, whereas others maintain relevance more gradually over longer time spans.
> > * For WIKI (yearly), the learned α values are consistently larger than in the other datasets across all window sizes. The values of λ are also generally higher, leading to steeper decay curves.
> >
> > In summary, CountTRuCoLa does not converge to a single temporal pattern. Instead, it learns a diverse set of curves, with substantial variation in parameters.
> >
> > Our results provide no evidence that recency or frequency parameters are transferable across datasets with different temporal granularities. Rather, the fitted parameters adapt to the underlying time scale: coarser datasets tend to learn sharper recency effects and flatter frequency curves, while finer-grained datasets show a wider spectrum of decay rates and frequency patterns. The learned parameters remain dataset-specific, and even rule-specific rather than universal.
> >
> > Please refer to the paper for more details.
> >
> >
> > ### Q5 (runtime):
> > We report a detailed analysis of runtimes in Appendix A8.3, Table 8. We specified runtimes for different datasets and the different subtasks as creation of examples, parameter learning (i.e. fitting), and rule application. Table 9 and 10 contain some information about other models and their runtimes, which have been executed on GPUs. These results indicate that our approach, which requires only CPU resources, achieves lower runtimes than GPU-accelerated baselines. However, we are aware that these results are not directly comparable, which we explained in Section A8.3.
> >
> > Do we use available CPU infrastructure efficiently?
> > Currently, CountTRuCoLa uses multithreading only in the rule application phase and the reported runtimes were conducted in a setting where 20 cores have been used. So far, we have implemented mltithreading only for the rule application phase, because the results indicate that it is for most datasets the most costly phase in terms of runtime. However, it would also be possible to implement multithreaded versions for the creation of examples or for the parameter learning, as any of these tasks can be conducted independently for each relation in the dataset.

---

> > > ### Author Response · Authors · 2025-11-28
> > > **Official Comment by Authors**
> > >
> > > We sincerely thank you again for reviewing our paper. With the discussion period nearing its end, we would like to confirm whether our responses have adequately addressed your concerns. Please feel free to reach out if you have any additional questions. We are more than willing to provide further explanations.

---

### Author Response · Authors · 2025-11-21
**Author Response to All Reviewers**

We thank the reviewers for their insightful reviews.

We are glad to see that the reviewers liked the
* Interpretability and transparency of the model `(Qwtd, 2EQa, jc2c)` as well as the explicit explanations `(2EQa)` and the explainer tool `(Qwtd)`
* Fully interpretable confidence functions with clear recency and frequency components and intuitive parameters `(Qwtd, 2EQa)`
* Comprehensive empirical evaluation across nine datasets, including ablations `(Qwtd, 2EQa, jc2c)` and reproducibility due to code, datasets, and detailed appendices `(Qwtd)`
* Competitive accuracy and scalability, even with a simple, CPU-only model `(Qwtd, jc2c)`


We appreciate the constructive feedback from the reviewers to help improve our paper. We integrated the feedback and uploaded a new version of our paper. Changes are highlighted in blue.

Based on the feedback, we added the following changes:
* A motivation for our choice of the recency function $f$ (Section 4.2) `(2EQa, W3)`
* Clarification of the term Interpretability (Section 4.4). In this context, we also replaced “explainable” with “interpretable” in the title and text. `(Qwtd, W1)`
* Ablation study on learning confidence-function parameters (Section 5.2, Table 2(c)) `(2EQa, W4)` and `(jc2c, W2)`
* Hyperparameter sensitivity (Appendix A.7, Figure 3 Effect of selected hyperparameters on the test MRR and Table 6 Comparison to Default Hyperparameters) `(Qwtd, W4, Q2)` and `(2EQa, W4)`
* Rule Aggregation Analysis, Analysis of rule confidence (A.8.1, Figure 4 Analysis of Over- and Underestimation in Confidence Bins and Table 7 Analysis of Over- and Underestimation in Top-1 Predictions) `(Qwtd, W1, W4, Q2)`
* Analysis of Parameter Sensitivity to Learn Window Size (A.8.2, Figure 5 and Figure 6) `(Qwtd, Q4)`
* Rule Type Distribution and Score Distribution (A.8.3, Figure 7) `(Qwtd, Q3)`
* Hits@1 and Hits@3 results (A.8.5, Table 11 and Table 12) `(2EQa, W5)` and `(jc2c, W4)`
* Examples for Learned Recency Functions (A.10, Figure 8 and Figure 9)  `(2EQa, W3)` and `(jc2c, W2)`

**Please note that we changed the title in our new submission, as explained above, as a reaction to the reviewers comments. Unfortunately, it was not possible to change the title here in openreview, even though the guidelines say that it is possible.**

Further, we address the comments of each reviewer individually below.

---

### Note · Authors · 2025-12-17

**Comment:**

We would like to withdraw our paper. We did not receive a response to our rebuttal despite the significant effort that we invested. Due to the early stopping of the review process, further interaction with reviewers was no longer possible.

**Withdrawal Confirmation:**

I have read and agree with the venue's withdrawal policy on behalf of myself and my co-authors.